# Synaptic density affects clinical severity via network dysfunction in syndromes associated with frontotemporal lobar degeneration

David J. Whiteside [1,2] ✉, Negin Holland[1,2], Kamen A. Tsvetanov[3], Elijah Mak[4], Maura Malpetti [1,2], George Savulich [4], P. Simon Jones [1], Michelle Naessens[3], Matthew A. Rouse[3], Tim D. Fryer[1,5], Young T. Hong[1,5], Franklin I. Aigbirhio[1,5], Eoin Mulroy[6], Kailash P. Bhatia[6], Timothy Rittman [1,2], John T. O'Brien [2,4,7] & James B. Rowe [1,2,3,7]

There is extensive synaptic loss from frontotemporal lobar degeneration, in preclinical models and human in vivo and post mortem studies. Understanding the consequences of synaptic loss for network function is important to support translational models and guide future therapeutic strategies. To examine this relationship, we recruited 55 participants with syndromes associated with frontotemporal lobar degeneration and 24 healthy controls. We measured synaptic density with positron emission tomography using the radioligand [$^{11}$C]UCB-J, which binds to the presynaptic vesicle glycoprotein SV2A, neurite dispersion with diffusion magnetic resonance imaging, and network function with task-free magnetic resonance imaging functional connectivity. Synaptic density and neurite dispersion in patients was associated with reduced connectivity beyond atrophy. Functional connectivity moderated the relationship between synaptic density and clinical severity. Our findings confirm the importance of synaptic loss in frontotemporal lobar degeneration syndromes, and the resulting effect on behaviour as a function of abnormal connectivity.

Frontotemporal lobar degeneration (FTLD) pathologies cause heterogenous syndromes with partially overlapping clinical features and highly variable correlations between neuropathology and phenotypic expression[1–4]. These conditions are associated with early loss of functional independence, considerable care burden and reduced life expectancy[5,6]. There is a pressing need for new therapeutic interventions, based on better characterisation and in vivo analysis of the pathogenic cascade leading from the molecular root causes to clinical presentation and progression[7,8]. This cascade includes severe synaptic loss[9,10], which integrates the toxic effects of protein aggregation and inflammation[11–14].

We propose that the severe synaptic loss arising from frontotemporal lobar degeneration would impair local and long-range functional connectivity, and thereby cognitive and behavioural change. We test this hypothesis with three clinical syndromes associated with different types of frontotemporal lobar degeneration; progressive supranuclear palsy (PSP), corticobasal syndrome (CBS)

[1]Department of Clinical Neurosciences, University of Cambridge, Cambridge, UK. [2]Cambridge University Hospitals NHS Foundation Trust, Cambridge, UK. [3]Medical Research Council Cognition and Brain Sciences Unit, University of Cambridge, Cambridge, UK. [4]Department of Psychiatry, University of Cambridge, Cambridge, UK. [5]Wolfson Brain Imaging Centre, University of Cambridge, Cambridge, UK. [6]UCL Queen Square Institute of Neurology, University College London, London WC1N 3BG, UK. [7]These authors jointly supervised this work: John T. O'Brien, James B. Rowe. ✉e-mail: djw216@cam.ac.uk

and behavioural variant frontotemporal dementia (bvFTD). These phenotypic entities show molecular heterogeneity, with 3-repeat and 4-repeat tauopathies and TDP-43 pathology. By studying these distinct syndromes from the FTLD-spectrum, we capture variations in anatomical distribution of pathology to better assess its consequence for connectivity and clinical severity.

The radioligand [11C]UCB-J quantifies synaptic density through selective binding to the presynaptic vesicle glycoprotein 2 A (SV2A). When used in positron emission tomography (PET) imaging, it confirms the reduction of synaptic density in bvFTD[10], CBS and PSP[9,15]. Reductions in [11C]UCB-J non-displaceable binding potential ($BP_{ND}$), a metric of binding site density, correlate with clinical severity and are observed in the distributions also found in post mortem studies[16,17]. [11C]UCB-J $BP_{ND}$ is primarily a measure of synaptic density rather than synaptic function[18] and is directly related to changes in cortical neurophysiological generators in FTLD[19]. Disruption to functional connectivity and network integration in FTLD-disorders aligns closely with symptom onset and progression[20,21], although uncertainty exists as to the cellular and molecular processes that determine functional connectivity as measured from resting-state functional MRI (fMRI)[22,23].

In this work we show that synaptic density is related to large-scale brain connectivity and network organisation. We undertook a multimodal neuroimaging study to combine [11C]UCB-J PET with resting state functional MRI in participants with three FTLD-associated disorders and similarly-aged healthy controls. We also used neurite orientation dispersion and density imaging (NODDI, a diffusion magnetic resonance imaging method[24], to provide complementary evidence for the role of synaptic health on connectivity. We quantified local functional connectivity using the graph metric of weighted degree. We show that local synaptic loss is associated with reduced connectivity, and that reductions in synaptic density explain connectivity loss that is not accounted for by atrophy. To examine syndrome-specific differences in the effect of synaptic loss, we reduced data dimensionality by independent component analysis. We found that reduced synaptic density was associated with connectivity loss in such a way as moderates and explains additional variance in the individual differences in cognition. We further explored the effect of synaptic loss on connectivity, distinguishing the effect of synaptic loss in a region versus the region to which it is connected.

## Results
### Clinical characteristics of participants
Demographic details and clinical characteristics of participants are set out in Table 1 and Supplementary Table 1. No significant group differences were observed for age or sex. Clinical and neuropsychological assessments showed impairment in all patient groups, with expected higher average scores on the Cambridge Behavioural Inventory-Revised (CBI-R) in bvFTD and greater impairment on the PSP-rating-scale (PSPRS) in PSP than bvFTD. Participants with bvFTD had increased in-scanner motion during fMRI acquisition, and therefore we included a metric of motion as a covariate of no interest throughout.

### Group differences in synaptic density, neurite dispersion, connectivity, and grey matter
There were widespread significant reductions in [11C]UCB-J $BP_{ND}$, the orientation dispersion index (ODI) determined with NODDI, grey matter volume, and weighted degree in patients compared to control participants in cortical and subcortical regions. The largest effect sizes were seen for [11C]UCB-J $BP_{ND}$ (Supplementary Fig. 1). The distribution of group differences were similar for [11C]UCB-J $BP_{ND}$ and ODI, with large between-group changes in these modalities spanning the frontal lobe, cingulate, insula, and basal ganglia. For grey matter volumes, there were prominent decreases in the frontal lobe and basal ganglia. The largest reductions in weighted degree in patients compared to controls were observed for peri-rolandic regions and the cerebellar dentate. The extensive differences between controls and patients in [11C]UCB-J $BP_{ND}$ were observed even after local volume was included as a covariate, in both regional and voxelwise analyses (Supplementary Fig. 2).

For weighted degree and [11C]UCB-J $BP_{ND}$ there were no regional differences between-patient groups, after correcting for multiple comparisons. Uncorrected reductions for [11C]UCB-J $BP_{ND}$ were greater in bvFTD than CBS and PSP, in the frontal lobe, insula, cingulate, and anterior temporal lobe; and for PSP when compared to CBS in the pallidum (Supplementary Table 2). Uncorrected reductions in weighted degree in bvFTD versus CBS (Supplementary Table 3) were observed in the anterior temporal lobe and substantia nigra. Significantly lower ODI values for bvFTD than PSP and CBS were found for widely distributed regions across the brain (Supplementary Table 4), with the largest effect sizes in the frontal and temporal lobe and insula. There were smaller ODI values for PSP than CBS for subcortical structures. Reduced grey matter volumes in bvFTD compared to CBS and PSP (Supplementary Table 5) were found in the frontal and temporal lobes, the cingulate, insula, and caudate, with reductions for CBS compared to PSP in the precentral gyri. We found reduced total volumes for PSP compared to CBS in the midbrain and cerebellum dentate.

## Table 1 | Demographic and clinical characteristics for participants

| | Control | PSP | CBS | bvFTD | Statistic (F/$\chi 2$) | P | Post-hoc tests |
|---|---|---|---|---|---|---|---|
| N | 24 | 29 | 16 | 10 | | | |
| Age at fMRI | 70.0 (8.4) | 70.8 (8.4) | 67.1 (5.7) | 65.0 (9.1) | 1.6 | 0.19 | |
| Sex (M/F) | 16/8 | 15/14 | 7/9 | 8/2 | 4.6 | 0.21 | |
| Mean DVARS | 5.0 (0.4) | 5.2 (0.6) | 4.9 (0.4) | 5.9 (0.8) | 7.8 | 0.0001 | bvFTD > Control $p = 0.0007$<br>bvFTD > PSP $p = 0.0075$<br>bvFTD > CBS $p = 0.0001$ |
| ACE-R | 95.8 (2.6) | 78.6 (13.4) | 77.8 (16.9) | 63.1 (29.0) | 12.5 | $1 \times 10^{-6}$ | bvFTD <Control $p < 0.0001$<br>bvFTD <PSP $p = 0.033$<br>PSP <Control $p = 0.0007$<br>CBS <Control $p = 0.003$ |
| PSPRS | – | 34.0 (11.1) | 27.2 (11.1) | 17.6 (11.2) | 7.3 | 0.002 | PSP > bvFTD $p = 0.002$ |
| CBI-R | – | 53.2 (34.3) | 37.7 (19.8) | 86.9 (34.5) | 7.9 | 0.0009 | bvFTD > CBS $p = 0.0007$<br>bvFTD > PSP $p = 0.012$ |

Scores are mean (Sd). Categorical and continuous variables were compared across groups using chi-squared tests and one-way analysis of variance respectively. Post-hoc pairwise comparisons used two-sided testing with Tukey method to adjust P values. *fMRI* functional magnetic resonance imaging, *DVARS* spatial standard deviation of successive images, *ACE-R* Addenbrooke's Cognitive Examination-Revised, *PSPRS* Progressive Supranuclear Palsy Rating Scale, *CBI-R* Cambridge Behavioural Inventory Revised.

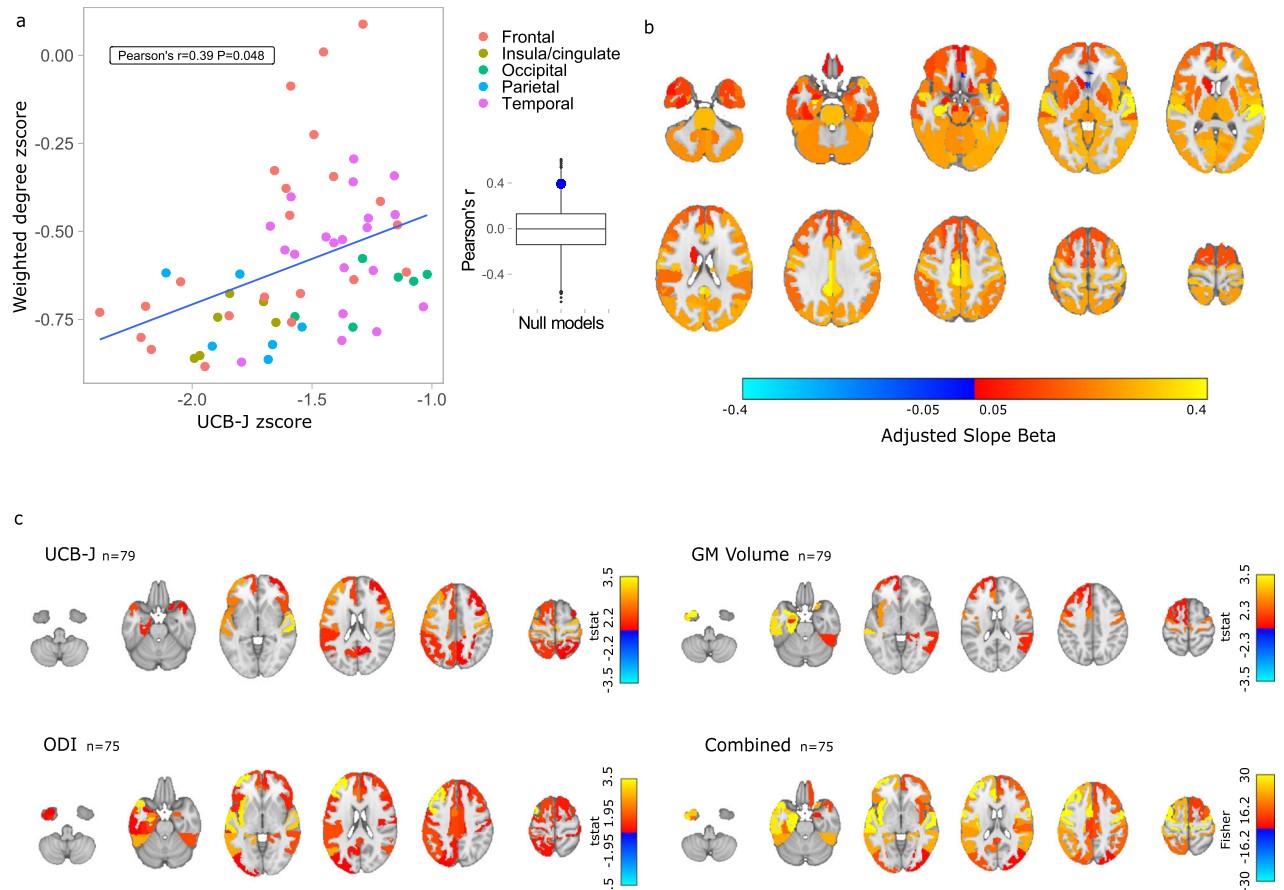

**Fig. 1 | Weighted degree and synaptic density in frontotemporal lobar degeneration syndromes. a** Mean [$^{11}$C]UCB-J binding potential (BP$_{ND}$) in cortical regions in patients ($n = 55$) is associated with mean weighted degree, with z scores calculated relative to control values. P value calculated using two-sided spatial-autocorrelation preserving permutation testing. The distribution of null models is shown in the box-plot, with median, inter-quartile range, and whiskers extending from the hinge to the largest/smallest point at most 1.5 times the interquartile range from the hinge. **b** Variation in strength of the weighted degree-[$^{11}$C]UCB-J BP$_{ND}$ relationship in patients ($n = 55$) derived from the effect of [$^{11}$C]UCB-J BP$_{ND}$ slope

within each region, adjusted by the [$^{11}$C]UCB-J BP$_{ND}$ fixed effect. Larger standardised estimates were observed across the cortex and away from inferior frontal, anterior temporal and striatal regions. **c** Regions with a relationship with weighted degree in a general linear model for all participants for [$^{11}$C]UCB-J BP$_{ND}$, orientation dispersion index (ODI), grey matter (GM) volumes, and for the three modalities combined using non-parametric combination testing with correction across contrasts. Regions shown are significant with FDR-adjusted p-values < 0.05. Source data are provided as a Source Data file.

## Weighted degree is related to synaptic density in patients but not control participants

We calculated regional [$^{11}$C]UCB-J BP$_{ND}$ and weighted degree z scores in patients standardised to control data. For cortical regions, mean weighted degree z scores correlated with mean [$^{11}$C]UCB-J BP$_{ND}$ z scores (Pearson's r = 0.39 P = 0.048, Fig. 1a). Parcels in the subgenual frontal cortex and subcallosal area lay off the main slope. The relationship was present even without these outliers, with a significant correlation in non-frontal cortical regions (Pearson's r = 0.43 P = 0.016). There was no such relationship in subcortical regions (Pearson's r = −0.1 P = 0.78).

We fitted linear mixed-effects models to account for individual variation in the regional relationship between [$^{11}$C]UCB-J BP$_{ND}$ and weighted degree. Inclusion of an effect of [$^{11}$C]UCB-J BP$_{ND}$ slope within each region improved model fit ($\chi^2 = 42$, $P = 8 \times 10^{-11}$). Weighted degree was associated with [$^{11}$C]UCB-J BP$_{ND}$ in patients (Standardised Beta 0.19, $P = 6 \times 10^{-11}$) but not in control participants (Standardised Beta −0.01 P = 0.90). The group-by-[$^{11}$C]UCB-J BP$_{ND}$ interaction in a refitted model with all participants was significant ([$^{11}$C]UCB-J BP$_{ND}$*Group Standardised Beta 0.067, $P = 0.0008$), with unchanged effect size and significance with scanning interval included as a covariate of no interest. In patients, assessing the [$^{11}$C]UCB-J BP$_{ND}$ effect within the region showed higher standardised betas for the weighted degree-[$^{11}$C]

UCB-J BP$_{ND}$ relationships in temporal, parietal, cingulate, and superior frontal regions (Fig. 1b). The relationship between [$^{11}$C]UCB-J BP$_{ND}$ and weighted degree remained significant (Standardised Beta 0.18, $P = 1 \times 10^{-9}$, Supplementary Table 6) with the inclusion of regional grey matter volumes and total intracranial volume, and was also observed in each FTLD syndrome individually (bvFTD Standardised Beta 0.2 $P = 1 \times 10^{-5}$; CBS Standardised Beta 0.12 P = 0.007; PSP Standardised Beta 0.23, $P = 4 \times 10^{-9}$). Using the non-reparcellated Hammersmith atlas and the Brainnetome parcellation we similarly found an association between synaptic density and weighted degree, and that this relationship was stronger in patients than control participants (Supplementary Results).

To further explore the effect of synaptic health on connectivity, we considered the complementary metric ODI. In linear mixed-effects models, regional [$^{11}$C]UCB-J BP$_{ND}$ values were related to regional ODI in patients (Standardised Beta 0.54 $P < 2 \times 10^{-16}$) and controls (Standardised Beta 0.39 $P < 2 \times 10^{-16}$). ODI values were significantly associated with a regional weighted degree in patients (Standardised Beta 0.056 P = 0.007) but not in controls (Standardised Beta 0.028 P = 0.39). The group-by ODI interaction was not significant (ODI*Group Standardised Beta 0.014, P = 0.51). The relationship between ODI and weighted degree persisted with grey matter volumes and total intracranial volume as covariates (Standardised Beta 0.043 P = 0.037).

For each cortical region, we tested the relationship between weighted degree and [$^{11}$C]UCB-J BP$_{ND}$, ODI, and grey matter volume across participants. The measures of synaptic health, [$^{11}$C]UCB-J BP$_{ND}$ and ODI, were significantly associated with weighted degree in more regions (Fig. 1c, Supplementary Fig. 3) than was grey matter volume. Combining modalities with non-parametric combination testing, including adjustment for the number of contrasts, resulted in the most widespread significant relationships with weighted degree. We further tested if regional [$^{11}$C]UCB-J BP$_{ND}$ was related to weighted degree with grey matter atrophy and total intracranial volume included as covariates in the general linear model. We found post-correction significant relationships in the right orbitofrontal gyrus and bilateral superior temporal gyri (Supplementary Fig. 4).

### An independent component analysis shows regional variation in synaptic density by clinical syndrome

To investigate how variation in synaptic density influences functional connectivity and cognition, we first performed an independent component analysis on concatenated participant [$^{11}$C]UCB-J BP$_{ND}$ partial volume corrected maps. One component was discarded as it was not robust to removal of an outlier (Grubb's test G 5.0 $P = 3 \times 10^{-6}$). Two further components were not taken forward for further analysis as they incorporated regions particularly prone to artefact. Six of the remaining components differed between groups (Fig. 2a, b, Supplementary Fig. 5): component 1 covering striatal regions (F(3,73) = 12.9 FDR $P = 1 \times 10^{-6}$; post-hoc Tukey bvFTD <Control $P = 0.0001$, CBS <Control $P = 0.0015$, PSP <Control $P < 0.0001$); component 2 covering the medial parietal lobe and adjacent parts of the frontal lobe (F(3,73) = = 5.8 FDR $P = 0.002$; post-hoc Tukey bvFTD <Control $P = 0.018$, bvFTD <PSP $P = 0.021$, CBS <Control $P = 0.029$, CBS < PSP $P = 0.024$); component 3 with spatial extent incorporating left frontoparietal regions (F(3,73) = 13.5 FDR $P = 1 \times 10^{-6}$; post-hoc Tukey bvFTD <Control $P < 0.0001$, bvFTD <CBS $P = 0.003$, bvFTD <PSP $P = 0.010$; PSP <Control $P = 0.0014$); component 4 with highest values in the anterior cingulate gyrus and insular cortex (F(3,73) = 26.8 FDR $P = 6 \times 10^{-11}$; post-hoc Tukey bvFTD <Control $P < 0.0001$, bvFTD <CBS $P < 0.0001$, bvFTD <PSP $P < 0.0001$, PSP <Control $P = 0.002$); component 5 covering the posterior cingulate and right peri-sylvian regions (F(3,73) = 9.9 FDR $P = 0.00002$; post-hoc Tukey bvFTD <Control $P < 0.0001$, bvFTD <CBS $P = 0.029$, PSP <Control $P = 0.003$); and component 6 with peak values in the left lateral frontal lobe and left peri-sylvian regions (F(3,73) = 12.7 FDR $P = 1 \times 10^{-6}$; post-hoc Tukey bvFTD <Control $P = 0.0001$, CBS<Control $P = 0.015$, PSP <Control $P < 0.0001$). Component identification was robust to alternative model order choices (9-14 components, mean spatial cross-correlation of matched component: component 1 0.79, component 2 0.97, component 3 0.88, component 4 0.89, component 5 0.95, component 6 0.93) and for components 2-6 to partial volume correction (uncorrected maps with model order 10, cross-correlation of matched component 0.85-0.95). Using uncorrected maps, component 1 was incorporated into a more spatially extensive component including frontal and parietal cortical regions.

We repeated source-based synaptometry following regression of grey matter tissue probability maps from the partial volume corrected [$^{11}$C]UCB-J BP$_{ND}$ maps. We found participant loadings were strongly correlated for the matched components, with a highly similar distribution of between-group differences as noted in our primary analysis (Supplementary Results, Supplementary Fig. 6).

### The functional connectivity of a region is associated with its synaptic density, both at the site of synaptic loss and remotely from it

Dual regression identified participant-specific patterns of spatial covariance to each [$^{11}$C]UCB-J BP$_{ND}$ component map. Participant component-specific functional connectivity maps were taken forward to assess for voxel-wise associations with [$^{11}$C]UCB-J BP$_{ND}$ component loadings in a general linear model with permutation testing. There were significant associations between increased [$^{11}$C]UCB-J BP$_{ND}$ loadings and increased functional connectivity for five of the components (Fig. 2b, $P < 0.05$ with family-wise error correction across voxels). We found that connectivity differences were observed both at the site of synaptic loss and remotely from it. For instance, greater [$^{11}$C]UCB-J BP$_{ND}$ loadings in a component (component 1) with high values in the bilateral striatum were associated with increased functional connectivity across the cortex.

We extracted participant component scores by taking the mean beta, normalised to the residual within-subject noise, from participants' component-specific connectivity maps within a mask defined as significant areas of control functional connectivity with each component. Five of the six components showed associations between [$^{11}$C]UCB-J BP$_{ND}$ loadings and functional connectivity score (Fig. 2c: component 1 Standardised Beta 0.4 FDR $P = 0.001$, component 2 Standardised Beta 0.29 FDR $P = 0.01$, component 3 Standardised Beta 0.27 FDR $P = 0.013$, component 4 Standardised Beta −0.17 FDR $P = 0.17$, component 5 Standardised Beta 0.43 FDR $P = 0.001$, component 6 Standardised Beta 0.4 FDR $P = 0.001$). This association remained significant for the same five components when patients alone were included in the model (component 1 Standardised Beta 0.35 FDR $P = 0.032$, component 2 Standardised Beta 0.27 FDR $P = 0.039$, component 3 Standardised Beta 0.28 FDR $P = 0.038$, component 4 Standardised Beta −0.13 FDR $P = 0.38$, component 5 Standardised Beta 0.37 FDR $P = 0.028$, component 6 Standardised Beta 0.36 FDR $P = 0.028$).

### Functional connectivity adds to and moderates the explanatory effect of UCB-J BP$_{ND}$ on clinical severity

We tested whether the inclusion of connectivity component scores to [$^{11}$C]UCB-J BP$_{ND}$ loadings improved modeling of clinical severity as measured by the ACE-R and PSPRS. Individual functional component scores were not significant predictors of either the ACE-R or the PSPRS, whereas [$^{11}$C]UCB-J BP$_{ND}$ component 2 (Standardised Beta 0.44 FDR $P = 0.002$), [$^{11}$C]UCB-J BP$_{ND}$ component 3 (Standardised Beta 0.49 FDR $P = 0.001$), UCB-J BP$_{ND}$ component 4 (Standardised Beta 0.64 FDR $P = 6 \times 10^{-6}$), and UCB-J BP$_{ND}$ component 6 (Standardised Beta 0.47 FDR $P = 0.0009$) were significantly associated with total ACE-R. Lower loadings on striatal [$^{11}$C]UCB-J BP$_{ND}$ component 1 were significantly associated with higher PSPRS scores (Standardised Beta −0.50 FDR $P = 0.001$). Stepwise regression using Bayesian Information Criteria determined whether combinations of predictors improved model fit. The winning model for ACE-R included [$^{11}$C]UCB-J BP$_{ND}$ component 2 and the fMRI-[$^{11}$C]UCB-J BP$_{ND}$ interaction terms for components 4 and 5 (Fig. 3a and Supplementary Table 7), suggesting that greater cognitive impairment is associated with cortical synaptic loss, moderated by frontal and insular connectivity. For both components 4 and 5, we found that the relationship between [$^{11}$C]UCB-J BP$_{ND}$ loadings and ACE-R was stronger in those with higher connectivity scores. The winning model for the PSPRS included the striatal-[$^{11}$C]UCB-J BP$_{ND}$ component 1 and the fMRI-[$^{11}$C]UCB-J BP$_{ND}$ interaction term for component 5 (Fig. 3b and Supplementary Table 8). We found that the relationship between UCB-J BP$_{ND}$ loading and PSPRS was only present in those with higher connectivity scores.

In summary, reductions in synaptic density were associated with reduced functional connectivity, and that connectivity both adds to and moderates the explanatory effect of synaptic density on clinical severity.

### Spatially remote functionally connected brain regions can moderate the effect of synaptic density on clinical severity

To test the effects of connectivity remote from synaptic loss, we considered connectivity scores from participants' component connectivity maps within connected regions but outside the [$^{11}$C]UCB-J BP$_{ND}$

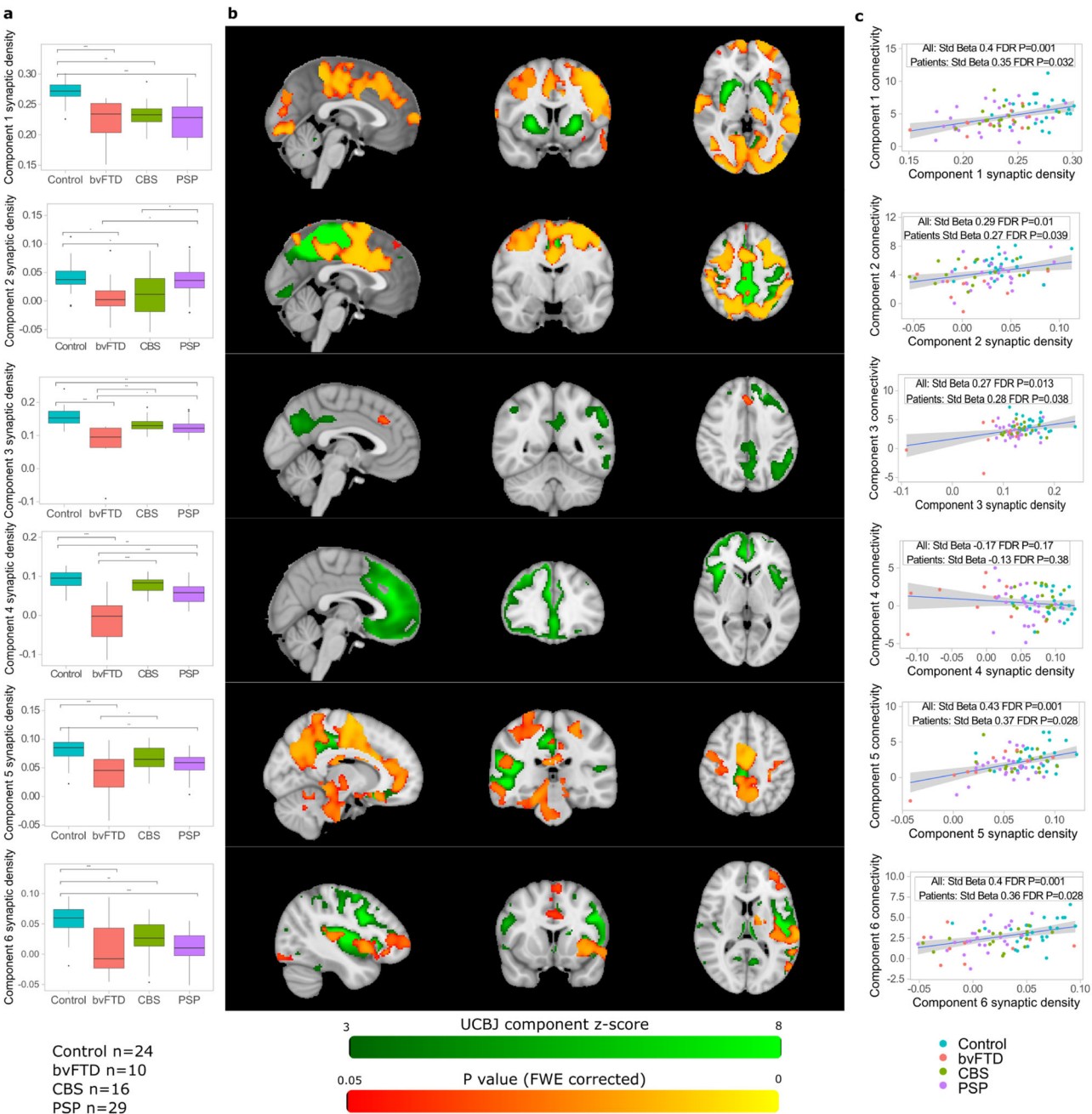

**Fig. 2 | Spatial variation in synaptic density and associated connectivity loss.**
**a** [^11C]UCB-J BP$_{ND}$ independent component analysis loadings by group for the six components that show differential expression in neurodegeneration in a linear model with false discovery rate correction across multiple comparison. Box and whisker plots show median, inter-quartile range, with whiskers extending from the hinge to the largest/smallest point at most 1.5 times the interquartile range from the hinge. Significance brackets (* $p < 0.05$, ** $p < 0.01$, *** $p < 0.001$) represent $P$ values from Tukey adjusted post-hoc pairwise comparisons. Component 1 bvFTD <Control $P = 0.0001$, CBS <Control $P = 0.0015$, PSP <Control $P < 0.0001$; component 2 bvFTD <Control $P = 0.018$, bvFTD <PSP $P = 0.021$, CBS <Control $P = 0.029$, CBS < PSP $P = 0.024$; component 3 bvFTD <Control $P < 0.0001$, bvFTD <CBS $P = 0.003$, bvFTD <PSP $P = 0.010$; PSP <Control $P = 0.0014$; component 4 bvFTD <Control $P < 0.0001$, bvFTD <CBS $P < 0.0001$, bvFTD <PSP $P < 0.0001$, PSP <Control $P = 0.002$; component 5 bvFTD <Control $P < 0.0001$, bvFTD <CBS $P = 0.029$, PSP <Control $P = 0.003$; component 6 bvFTD <Control $P = 0.0001$, CBS<Control $P = 0.015$, PSP <Control $P < 0.0001$. **b** [^11C]UCB-J BP$_{ND}$ component maps (in green) with areas of increased functional connectivity (in red-orange significantly associated with increased [^11C]UCB-J BP$_{ND}$ component loadings. $P$ values calculated using a one-sided permutation test with family-wise error correction across voxels. **c** Connectivity scores, derived from participant-specific functional connectivity maps per component, are associated with [^11C]UCB-J BP$_{ND}$ independent component analysis loadings. Standardised beta calculated from linear regression with two-sided testing, with false discovery rate adjusted $P$ values. The error bands represent the 95% confidence interval. bvFTD behavioural variant frontotemporal dementia, CBS corticobasal syndrome, PSP progressive supranuclear palsy, FWE family-wise error, FDR false discovery rate. Source data are provided as a Source Data file.

component. We repeated the model selection process above using connectivity scores outside the [^11C]UCB-J BP$_{ND}$ mask. We found that for the final model predicting ACE-R (Supplementary Table 9), the fMRI-[^11C]UCB-J BP$_{ND}$ interaction was significant for component 4 for the ACE-

R (Component 4 Interaction Std Beta 0.28, $P = 0.005$). The final model predicting PSPRS included only [^11C]UCB-J BP$_{ND}$ component 1 (Supplementary Table 10), although the fMRI-[^11C]UCB-J BP$_{ND}$ interaction was significant for component 5 with PSPRS as the dependent variable,

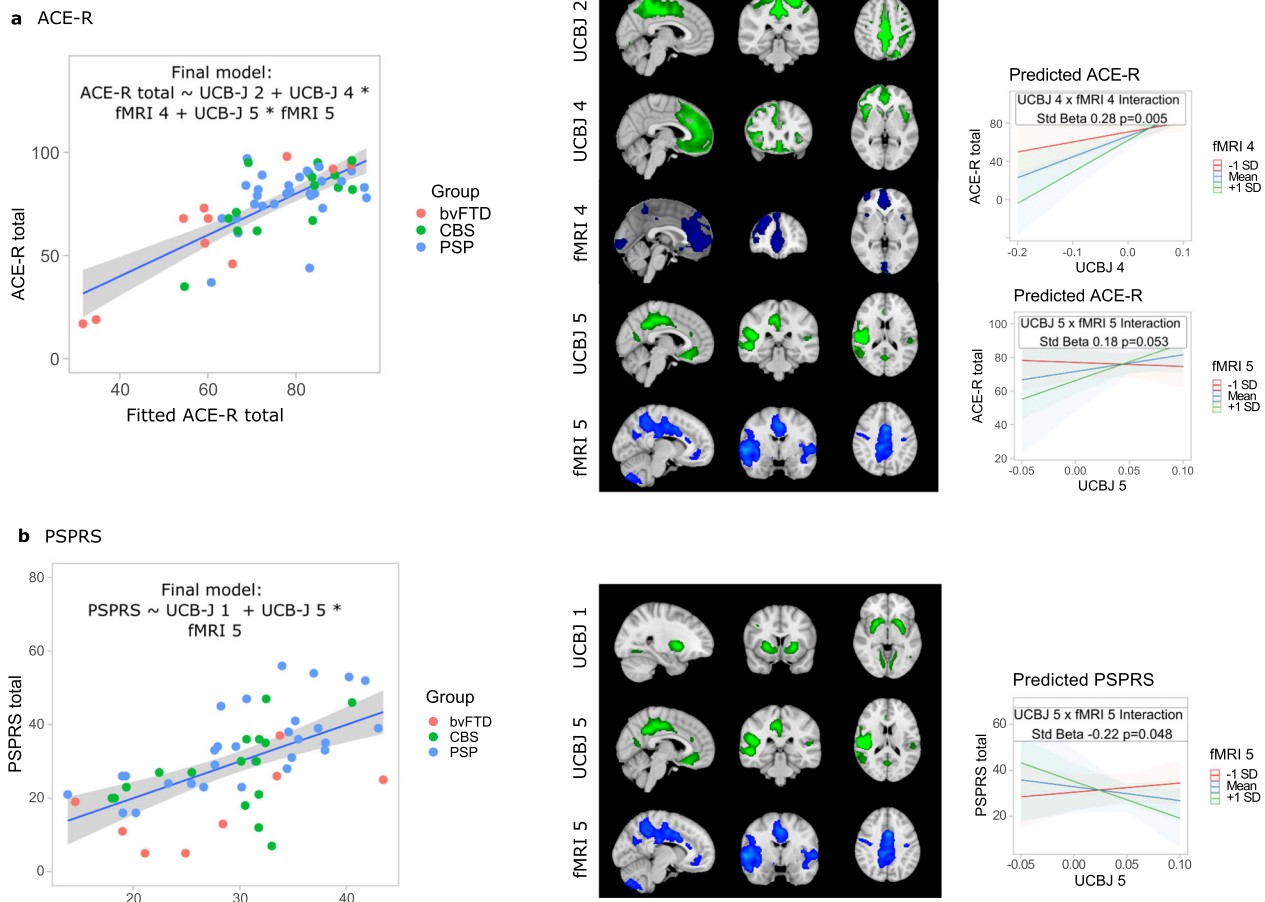

**Fig. 3 | Functional connectivity, synaptic density, and clinical severity.** [¹¹C] UCB-J BP$_{ND}$ components (in green) and distribution of functional connectivity (in blu**e**) for components included in final model selected using stepwise regression for **a** Addenbrooke's Cognitive Examination-Revised (ACE-R) and **b** Progressive Supranuclear Palsy Rating Scale (PSPRS). The relationship between the fitted final model and true scores is shown with a linear regression with error bands representing the 95% confidence interval. For the interactions included in the final models the relationship between [¹¹C]UCB-J BP$_{ND}$ component loading and clinical measure of severity was seen only in those with higher connectivity scores. Standardised betas and *P* values were calculated from two-sided testing from linear regression for the winning model, with the direction of the interaction terms illustrated with the marginal effect of the fMRI component with three groups (mean and +/- one standard deviation) and 95% confidence intervals. bvFTD: behavioural variant frontotemporal dementia, CBS: corticobasal syndrome, PSP progressive supranuclear palsy, fMRI functional magnetic resonance imaging. Source data are provided as a Source Data file.

again using component scores outside the [¹¹C]UCB-J BP$_{ND}$ component mask (Interaction Std Beta −0.31, *P* = 0.006). This supports the claim that remote but connected brain regions can moderate the effect of [¹¹C]UCB-J BP$_{ND}$ on cognition.

## Discussion

There are three principal results of this study. First, lower synaptic density as estimated by [¹¹C]UCB-J BP$_{ND}$ is associated with lower functional connectivity. Second, functional connectivity augments and moderates the relationship between synaptic density and clinical severity. Third, the [¹¹C]UCB-J BP$_{ND}$ estimates of synaptic density predict individual differences in connectivity over and above grey matter volume. The profound reductions in synaptic density in frontotemporal lobar degeneration occur in partially overlapping distributions, but with some disease-specific effects. The corresponding reductions in functional connectivity are observed both at the site of the synaptic loss and remotely from it: as one of the potential mechanisms underlying diaschisis, functional diaschisis, and the recent formalisation of connectomal diaschisis[25]. Leveraging information from multiple modalities allows us to estimate the relative contribution of individual determinants of brain functional connectivity.

Our study therefore provides important mechanistic insights into the multiple and interacting pathogenic processes of FTLD that result in diverse clinical syndromes.

Progressive supranuclear palsy, corticobasal syndrome and behavioural variant frontotemporal dementia are severely disabling progressive conditions that significantly reduce life expectancy[5,6]. To improve patient outcomes we require models of the human neurodegenerative pathogenesis as a bridge between preclinical studies and experimental medicines in people. This approach complements mesoscale mechanistic models of cortical function in frontotemporal lobar degeneration[19,26] and macroscale whole-brain models of neurodegeneration[27–29]. Such transitional markers may facilitate the selection and design of trials of potential disease-modifying agents[7].

Our primary finding of an association between lower synaptic density and reduced functional connectivity accords with the ubiquitous role of synaptic health and plasticity in generating neurophysiological connections, with synaptic change mediating information storage and contributing to learning[30,31]. In neurodegeneration, preclinical and neuropathological studies have shown that synaptic dysfunction and impaired plasticity are key determinants of impaired brain network organisation and cognitive dysfunction[14], and occur

before neuronal degeneration[32,33]. The in vivo associations between synaptic density and markers of connectivity derived are strengthened by the concordance across imaging and analytical methodologies.

The variances in functional connectivity explained by [¹¹C]UCB-J $BP_{ND}$ is independent of and in addition to that accounted for by grey matter atrophy, demonstrated through partial volume correction and by direct inclusion of grey matter volumes in the analysis models. This highlights the distinct contribution [¹¹C]UCB-J PET offers over and above T1-weighted MRI in capturing aspects of the neurodegenerative cascade. The histopathological processes that cause atrophy on structural imaging are incompletely characterised, with evidence for neuronal and synaptic loss, axon degeneration and cell death being important contributory factors[34,35]. Synaptic dysfunction and loss may occur without cell death or atrophy[12,32], in keeping with our finding of regions of reduced [¹¹C]UCB-J $BP_{ND}$ but without atrophy. Quantifying dendritic complexity through the diffusion imaging method NODDI provides converging evidence for the role of synaptic loss in network dysfunction. Since [¹¹C]UCB-J $BP_{ND}$ and ODI are independent predicters of connectivity and function, measuring synaptic density and neurite dispersion has the potential to improve our understanding of the individual role of synaptic loss in cognitive symptoms.

The distinct proteinopathies underlying frontotemporal lobar degeneration show overlap in clinical syndromes and anatomical distribution, while the same pathological entity can result in heterogeneous clinical presentations[1,4]. Despite the heterogeneity and pleiotropy of FTLD-disorders, synaptic loss and dysfunction is a common feature[36,37], associated with the direct and indirect toxic effects of multiple proteins' aggregates. We found patterns of synaptic density loss that were differentially expressed across the diagnostic labels in a manner in keeping with their typical neuropathological distributions described at post mortem[38,39], albeit with regional synaptic loss occurring in a continuum across participants. For instance, we found lower [¹¹C]UCB-J $BP_{ND}$ loadings in CBS than PSP for posterior components, with frontal synaptic loss in all conditions with the largest effect sizes in bvFTD. Synaptic dysfunction results in altered cognition and behaviour through loss of connectivity, with synaptic loss strongly predicting cognitive function and decline[40,41]. We have shown that connectivity improves prediction and moderates the effect of synaptic density on clinical severity, with functional connectivity loss both overlapping with and occurring remotely from sites of reduced synaptic density. These remote connectivity differences can themselves moderate the relationship between synaptic density and cognition.

Our work advances on recent studies of fMRI and SV2A radioligand PET in psychiatric disease[42] and neurodegeneration[43]. Our work differs in investigating a cohort with a heterogeneous distribution of synaptic loss, through adopting a whole-brain approach to demonstrate the anatomical effects of synaptic loss on connectivity, and by combining synaptic density, connectivity, and clinical severity in a single model. Nonetheless, there are several limitations to our study. Interpreting connectivity from resting state fMRI is made challenging by the low signal-to-noise ratio, the impact of movement and other artefact on measures of connectivity[44], the indirect neuronal interpretability of the BOLD signal[23] and the small effect sizes of brain-behaviour studies in a healthy population[45]. Yet it is a scalable and widely available modality with good spatial localisation that provides a neural correlate closely connected to behaviour, with network integrity closely aligned with symptom development in neurodegeneration[20,21]. It is therefore a potential outcome marker of the consequences of upstream neuropathological change. We found that the relationship between connectivity and [¹¹C]UCB-J $BP_{ND}$ showed regional heterogeneity. That the variance in functional connectivity was not fully explained by synaptic density raises the possibility for a role for other factors, such as the neurochemical deficits observed in FTLD[46]. Combining PET, fMRI, and structural imaging modalities with clinical and other data would allow individualised modelling of pathophysiological pathways with implications for developing personalised treatments[47,48].

We leverage the heterogeneity in FTLD to capture variation in the anatomical distribution of synaptic loss and explore its consequences but recognise that the small-moderate and varied group sizes mean that our study may be underpowered to investigate small differences between disease groups. Motion artifacts have a well-recognised confounding effect on functional connectivity[44,49], with higher motion observed in participants with bvFTD during resting state fMRI scanning. Despite careful pre-processing, including ICA denoising and wavelet despiking, together with inclusion of an in-scanner movement related parameter as a covariate of no interest in relevant models, it remains possible that residual artefact influences our findings. We acknowledge that longitudinal and interventional studies are required to demonstrate causality between synaptic dysfunction, brain network organisation and clinical severity, and cannot be assumed from the statistical associations observed here. Although SV2A expression is closely related to synaptic activity and function[50], [¹¹C]UCB-J $BP_{ND}$ is considered a measure of synaptic density rather than synaptic function[18]. Given that functional connectivity in fMRI is defined as a statistical dependency, with functional activation only indirectly related to neuronal activity, any inferences in our study made about synaptic function are necessarily implicit. Nonetheless our findings are as expected given the interplay between synaptic loss and dysfunction observed in preclinical studies[14]. We excluded participants with corticobasal syndrome and amyloid-positive PET imaging, but did not collect Alzheimer's disease biomarkers for patients with PSP and bvFTD. The clinicopathological correlation in PSP Richardson's syndrome is high[51,52], while Alzheimer's disease pathology can occur in behavioural variant FTD[53] and may alter synaptic density and connectivity. Lastly, we used reference tissue modelling rather than arterial blood sampling given the challenges of scanning in an FTLD-associated patient cohort. We have previously performed sensitivity analyses to ensure that group differences cannot be explained by any bias introduced through reference tissue selection[9].

To conclude, we report that reduced synaptic density in multiple frontotemporal lobar degeneration syndromes is associated with lower functional connectivity, with connectivity moderating the relationship between synaptic density and clinical severity. Synaptic density independently explains variance in connectivity beyond measuring atrophy from structural MRI. Our study provides in vivo support for preclinical findings and paves the way for individualised pathogenic models and personalised treatments.

## Methods
### Participants
29 participants with probable progressive supranuclear palsy, Richardson's syndrome[54], 16 participants with probable corticobasal syndrome and probable corticobasal degeneration[55], and 10 participants with behavioural variant Frontotemporal Dementia[56] were recruited from specialist clinics at the Cambridge Centre for Parkinson-plus, the Cambridge Centre for Frontotemporal Dementia, and National Hospital for Neurology and Neurosurgery at Queen Square, London[9,10]. 24 healthy volunteers were recruited from the UK National Institute for Health Research Join Dementia Research (JDR) register. Participants were initially screened via telephone, using the following exclusion criteria: a current or recent history of cancer within the last 5 years, concurrent use of the medication levetiracetam, any contra-indications to undergoing an MRI, a history of ischaemic or haemorrhagic stroke evident on the MRI from the clinic, and any severe physical illness or co-morbidity that could limit their ability to fully participate in the study. Twenty-three participants who passed initial screening were not included in this study due to (i) positive Alzheimer's biomarkers (all CBS, $n = 9$) or (ii) failure to complete all scanning

sessions (*n* = 14: Control *n* = 8, bvFTD *n* = 1, CBS *n* = 1, PSP *n* = 4). Participants sex was determined based on participant self-report and monitored during recruitment to ensure balance between patient groups and controls. The research protocol (18/EE/0059) was approved by the Cambridge Research Ethics Committee and the Administration of Radioactive Substances Advisory Committee. All participants provided written informed consent in accordance with the Declaration of Helsinki. Participants were compensated for any travel, food, and accommodation expenses incurred as part of the study.

Participants underwent study-specific clinical and neuropsychological assessment including the Mini-mental State Exam (MMSE[57], revised Addenbrooke's Cognitive Examination (ACE-R)[58], Progressive Supranuclear Palsy Rating Scale (PSPRS)[59], and the Cambridge Behavioural Inventory-Revised[60].

All participants underwent brain imaging with 3-Tesla MRI, including echo-planar imaging sequences sensitive to the blood-oxygen-level-dependent signal, and in a separate session PET scanning with [11C]UCB-J ((R)−1−((3-(methyl-11C)pyridin-4-yl)methyl)−4−(3,4,5-trifluorophenyl)pyr-rolidin-2-one[61]. All bar four participants (two PSP, one bvFTD, one control participant) additionally completed a diffusion sequence in the same session as echo-planar imaging (TE = 75.6 ms, TR = 2.4 s, slice thickness = 1.75 mm, 98 directions, 104 slices, bvals = 300, 1000, 2000). The median interval between scanning sessions was 58 days in patients (inter-quartile range 13-180 days) and 194 days in control participants (inter-quartile range 32-284 days). Participants with CBS also underwent amyloid PET using Pittsburgh compound B ([11C]PiB). Cortical standardised uptake value ratio (SUVR; 50−70 minutes post injection; whole cerebellum reference tissue was determined using the Centiloid Project methodology[62]. Only participants with corticobasal syndrome and a negative amyloid status, as characterised by a cortical [11C]PiB SUVR < 1.21 (obtained by converting the Centiloid cut-off of 19 to SUVR using the Centiloid-to-SUVR transformation[63]) are included in the analysis. In the absence of Alzheimer's disease, corticobasal degeneration (CBD) is the most common pathological finding in CBS, although we acknowledge other pathologies are possible[64].

## Data acquisition and processing

We have previously reported our protocol for [11C]UCB-J synthesis, data acquisition, image reconstruction and kinetic analysis[9]. Dynamic PET acquisition was performed on a GE SIGNA PET/MR (GE Healthcare, Waukesha, USA) for 90 minutes following [11C]UCB-J injection, with attenuation correction including the use of a multisubject atlas method[65] and improvements to the MRI brain coil component. Each emission image series was aligned to a T1-weighted MRI acquired during the same session (TE = 3.6 ms, TR = 9.2 ms, 192 sagittal slices, in-plane resolution 0.55 × 0.55 mm [interpolated to 1.0 × 1.0 mm]; slice thickness 1.0 mm). We derived a [11C]UCB-J BP$_{ND}$ map for each participant from a dynamic image series corrected for partial volume effects using the iterative Yang method[66]. For regional analysis we used a modified version of the n30r83 Hammersmith atlas (http://brain-development.org) including segmentation of brainstem and cerebellar structures, with the atlas non-rigidly registered to the T1-weighted MRI of each participant. Regions of interest were multiplied by a binary grey matter mask determined from the SPM12 (https://www.fil.ion.ucl.ac.uk/spm/software/spm12/) grey matter probability map smoothed to PET resolution and thresholded at >50%, unless masking eliminated the region in some subjects. For regional analysis, correction for cerebrospinal fluid partial volume error was applied to each image of the dynamic series. [11C]UCB-J binding potential (BP$_{ND}$) was calculated both at the regional and voxelwise level using a basis function implementation of the simplified reference tissue model[67], with centrum semiovale as the reference tissue[68]. We further calculated regional [11C]UCB-J BP$_{ND}$ for parcels of the Brainnetome Atlas[69] to test whether the results were independent of parcellation. For independent component analysis (see below), [11C]UCB-J BP$_{ND}$ maps were registered to the FSL MNI152 6$^{th}$ generation atlas using the warps and linear transforms created in spatial normalisation of the T1-weighted images using Advanced Normalization Tools (ANTs)[70] version 2.1.0. Normalised maps were smoothed with a 6 mm Gaussian kernel.

Functional MRI was performed with a 3-Tesla Siemens Prisma (Siemens Healthcare using echo-planar imaging sensitive to the blood-oxygen-level-dependent signal (TR 2.5 secs, TE 30 ms, whole brain acquisition, 3 x 3 x 3.5 mm voxels, 200 volumes). High resolution T1-weighted Magnetisation Prepared Rapid Gradient Echo (MPRAGE structural images (TR 2, TE 2.93 ms, voxel size 1.1 mm isotropic) were acquired during the same session. fMRI preprocessing followed the FSL pipeline[71] (using version 6.0.4), with the addition of wavelet despiking[72] given that motion artefact may be more common in patients with neurodegenerative diseases. T1 structural images were cropped to remove non-brain tissue followed by brain extraction using FSL's Brain Extraction Tool (BET). We then used FSL's FEAT with the following steps: motion correction using MCFLIRT; spatial smoothing using a Gaussian kernel of 5 mm full-width at half maximum; grand-mean intensity normalisation of the 4D dataset by a single multiplicative factor; and 100 Hz high-pass temporal filtering. Structured artefacts were removed using independent component analysis denoising using FSL's MELODIC together with FIX. FIX was hand-trained using a set of 20 subjects. Registration to high resolution structural and/or standard space images was carried out using FLIRT. Registration from high-resolution structural to MNI space was then further refined using FNIRT nonlinear registration. We did not use global signal regression. Wavelet despiking was used for further removal of motion artefact[72]. For dual regression analysis data was further smoothed with a 6 mm FWHM Gaussian kernel.

From volumetric T1-weighted MRI images grey matter volumes and total brainstem volumes were extracted for the same regions of the Hammersmith Atlas using SPM12 (version 7771) segmentation. Total intracranial volume was calculated via direct segmentation using Sequence Adaptive Multimodal SEGmentation[73]. We followed our previously reported pipeline for analysis of diffusion-weighted MRI to obtain the neurite orientation dispersion and density imaging (NODDI) metric orientation dispersion index (ODI)[15]. Non-brain tissue was removed from diffusion datasets using BET. Eddy currents and in-scanner head motion were corrected using FSL's eddy. We applied TOPUP to correct for correction of susceptibility induced distortions. Eddy was further used for quantitative identification of slices with signal loss, which were replaced by non-parametric predictions with the Gaussian process[74]. The b-matrix was reorientated by applying the rotation part of the affine transformation used during eddy correction[75]. ODI maps were derived using the Microstructural Diffusion Toolbox[76]. ODI maps were warped to MNI space using parameters from the spatial normalisation of the co-registered T1 image using ANTS. We then obtained regional ODI values for the modified Hammersmith Atlas within a grey matter mask using FSL's fslstats function.

## Weighted degree

Participants' preprocessed fMRI was parcellated using the modified Hammersmith atlas, with cortical regions masked with a grey matter mask. The straight gyrus was excluded given limited coverage in some participants. Given the variation in parcel size in the Hammersmith atlas, we sub-parcellated the masked Hammersmith atlas with 161 regions of approximately equal volume, such that each sub-parcel could be uniquely identified with an atlas region. Pearson correlations were calculated between nodes, followed by Fisher's r-to-Z transformation. Weighted degree was derived from association matrices using the Maybrain software (https://github.com/RittmanResearch/maybrain) and Networkx. We then calculated mean weighted degree

across the sub-parcellations for each Hammersmith atlas region to compare with regional [11C]UCB-J BP$_{ND}$.

## Comparing modalities by group

We compared regional values for [11C]UCB-J BP$_{ND}$, weighted degree, grey matter volumes/total brainstem volumes, and ODI between participants with FTLD-associated syndromes and controls in a linear model with age and sex as covariates (plus mean DVARS for weighted degree and total intracranial volume for volumetric measures). We further compared differences in the four modalities between patient groups in a linear model with the same covariates. For both analyses *P* values were adjusted for the false discovery rate across regions[77]. We additionally tested for differences in [11C]UCB-J BP$_{ND}$ with grey matter volume and total intracranial volume included as covariates of no interest in regional and voxelwise analysis (see Supplementary Methods).

## Weighted degree and synaptic density

We tested the spatial correlation between regional z scores for weighted degree and [11C]UCB-J BP$_{ND}$ using mean values and standard deviation from the control participants. *P*-values were calculated using a permutation test with 5000 spatial autocorrelation-preserving null models of the weighted degree parcellation[78], using the neuromaps toolbox[79].

To capture the effect of individual variability in the relationship between weighted degree and [11C]UCB-J BP$_{ND}$, we then derived mixed linear effects models for all patients using the lme4 package in R[80] with crossed random effects for region and participant and an effect of [11C]UCB-J BP$_{ND}$ slope within each region. We compared models using the anova function in R to ensure that the inclusion of a random slope for [11C]UCB-J BP$_{ND}$ per region improved model fit. We repeated the model with control participants alone, and for each clinical syndrome individually but without a random slope for [11C]UCB-J BP$_{ND}$ given convergence failures in the CBS and bvFTD groups. We tested the difference between patients and controls with a further model using all participants including a group by [11C]UCB-J BP$_{ND}$ interaction. Age at fMRI scan, sex and a marker of fMRI motion (mean DVARS)[44] were included as covariates of no interest. We also included a covariate denoting whether a region was cortical or subcortical, to ensure that the relationship between [11C]UCB-J BP$_{ND}$ and weighted degree was not driven by differences due to the average lower [11C]UCB-J BP$_{ND}$ in subcortical regions, and a group by cortical/subcortical interaction in the model with all participants. We performed a sensitivity analysis with scanning interval as an additional covariate. We further tested whether the inclusion of regional grey matter volume and total intracranial volume in the model in patients altered the [11C]UCB-J BP$_{ND}$-weighted degree relationship. We repeated models assessing the relationship between [11C]UCB-J BP$_{ND}$ and a weighted degree in patients and control participants using the Brainnetome parcellation and the non-reparcellated Hammersmith Atlas, with weighted degree calculated for regions of these atlases using the methodology outlined above.

To assess converging evidence for the role of synaptic health in determining functional connectivity, we considered the complementary NODDI derived metric ODI. We tested the relationship between weighted degree and ODI (and between [11C]UCB-J BP$_{ND}$ and ODI) in the mixed linear effects models outlined above. In controls, the model comparing ODI to [11C]UCB-J BP$_{ND}$ was fit without a slope for [11C]UCB-J BP$_{ND}$ in region due to model singularity.

To inform the contribution of different modalities to functional connectivity we compared weighted degree to [11C]UCB-J BP$_{ND}$, ODI, and grey matter volumes, across participants and within each cortical region separately. Relationships were tested in a general linear model with age at fMRI, sex, and mean DVARS as covariates, plus total intracranial volume for grey matter volumes. We used permutation testing using FSL's Permutation Analysis of Linear Models[81] (PALM

version alpha119) with 10000 permutations and correction for the false discovery rate across cortical regions. To test if the three modalities in combination would improve the prediction of weighted degree, we performed joint inference with non-parametric combination testing[82], with correction across the number of contrasts and the same covariates. In addition, we tested whether the relationships between weighted degree and [11C]UCB-J BP$_{ND}$ would persist with grey matter and total intracranial volume as covariates of no interest, both at regional and voxelwise (Supplementary Methods) levels.

## Source-based synaptometry

We then proceeded to independent component analysis (referred to here as source-based synaptometry, analogous to "volumetry") to identify a small number of statistically independent components capturing spatial variation in [11C]UCB-J BP$_{ND}$. Spatially concatenated [11C]UCB-J BP$_{ND}$ maps were submitted to source-based synaptometry using the GIFT toolbox[83,84] with a model order of 10, with regions with negative values in any participants masked out. Components were discarded if they represented artefact, captured regions known to be sensitive to artefact in fMRI acquisition or were driven by outliers identified using Grubbs' test[85] (i.e. the component did not persist with the outlier removed). Component loading values, which represent the degree to which an individual expresses a [11C]UCB-J BP$_{ND}$ component map, were taken forward to estimate association with connectivity providing they were differentially expressed by participants with neurodegenerative diseases and controls after correction for false discovery rate with $p < 0.05$. Six components satisfying these criteria were included. In the primary analysis presented here partial volume corrected [11C]UCB-J BP$_{ND}$ maps were used, with repeat source-based synaptometry with uncorrected maps performed to ensure robustness of spatial distribution of components to atrophy correction. Independent component analysis model order was chosen a priori, with additional analysis that components of interest with similar spatial distributions could be extracted at alternative model orders.

## Connectivity of [11C]UCB-J BP$_{ND}$ components

We then sought to investigate subject functional spatial covariance with the identified [11C]UCB-J BP$_{ND}$ components, using a seed-based dual regression approach[86,87]. In the first stage of dual regression, we regressed the selected [11C]UCB-J BP$_{ND}$ components maps into each participant's fMRI 4-dimensional dataset to give participants specific timecourses per component. These timecourses were taken to a second regression with the [11C]UCB-J BP$_{ND}$ component maps to obtain participant spatial maps per component, termed here component-specific functional connectivity maps. We assessed the association between [11C]UCB-J BP$_{ND}$ loadings values and each voxel of the component-specific functional connectivity maps in a general linear model with age, sex, and mean DVARS as covariates of no interest using threshold-free cluster enhancement with 5000 permutations using FSL's randomise tool[81] with family-wise error significance level $p < 0.05$. We then calculated the mean beta, normalised to the residual within-subject noise, for each participant's component-specific functional connectivity maps from the second stage of dual regression (within a mask defined as regions in controls showing significant mean connectivity with each [11C]UCB-J BP$_{ND}$ component map at family-wise error $p < 0.01$) using FSL's fslmeants function. We compared the association between [11C]UCB-J BP$_{ND}$ loadings and functional connectivity scores for each component in the whole group and in patients alone using the same covariates of no interest.

## Associations with clinical scores

We and others have previously shown that [11C]UCB-J BP$_{ND}$ values are strongly related to markers of clinical severity in neurodegenerative

diseases[10,88,89]. We sought to understand whether connectivity moderates this relationship and improves the prediction of total ACE-R and total PSPRS. We, therefore, performed model selection using stepwise regression with the Bayesian information criteria from a baseline model of:

$$ACE - R/PSPRS \sim [^{11}C]UCB - J\,component1*fMRI\,component1 + \dots$$
$$+ [^{11}C]UCB - J\,component6 * fMRI\,component6 + Age + Sex + Mean\,DVARS.$$

Age at fMRI, sex and mean DVARS were considered covariates of no interest and were not stepped out of the model.

### Remote connectivity differences

We proceeded to explore the effects on cognition of connectivity differences at the site of synaptic loss and of the region connected to it. To do this we thresholded and binarised the $[^{11}C]$UCB-J BP$_{ND}$ component maps beyond the 97.5th percentile point. We calculated connectivity scores, as outlined above, both within the component mask and within connected regions but outside the component mask. Then, on a post hoc basis, we repeated the stepwise regression for PSPRS and ACE-R using only the connectivity scores outside the $[^{11}C]$UCB-J BP$_{ND}$ component mask.

### Statistical analysis

We tested the relationships between $[^{11}C]$UCB-J BP$_{ND}$, ODI, functional connectivity, grey matter volume, and cognition in regional and voxelwise analyses. First, we tested whether regional $[^{11}C]$UCB-J BP$_{ND}$ and ODI explain regional variation in connectivity beyond that accounted for by grey matter volume. We then used voxel-wise $[^{11}C]$UCB-J BP$_{ND}$ maps to identify patterns of $[^{11}C]$UCB-J BP$_{ND}$ that were differentially expressed in neurodegeneration using an independent component analysis[76]. We identified participants-specific maps of functional spatial covariance with the $[^{11}C]$UCB-J components through seed-based dual regression. We tested whether variability in functional connectivity to regions showing group differences in synaptic density: a) improves modelling of clinical severity as assessed through the ACE-R and PSPRS; b) moderates the effect of $[^{11}C]$UCB-J BP$_{ND}$ differences on cognition. We further tested whether the effects of $[^{11}C]$UCB-J BP$_{ND}$ differences on cognition were influenced by connectivity remote from synaptic loss. Statistical analyses and visualisation were performed in R[90] version 4.1.2, unless stated otherwise.

### Reporting summary

Further information on research design is available in the Nature Portfolio Reporting Summary linked to this article.

## Data availability

Derived imaging data, including those required to generate Figs. 1–3 and supplementary figures, are provided in the Source Data file. These data have also been deposited in a Figshare repository (https://doi.org/10.6084/m9.figshare.24188580)[91]. Access to raw imaging data and linked clinical data may be available via an agreed third party Secure Research Environment or by transfer under a material/data transfer agreement, subject to conditions required to comply with participant consent and data protection regulations. Requests should be addressed to the senior author in the first instance. Initial responses to queries or requests for data, determining the need for any material/data transfer agreement, will be provided within a month of a request being made. The n30r83 Hammersmith atlas was modified for regional analysis and is publicly available at http://brain-development.org. The Brainnetome Atlas (https://atlas.brainnetome.org/) used for ensuring results are independent of parcellation in this study is publicly available. Source data are provided with this paper.

## Code availability

The Maybrain toolbox, a Python package for analysis and visualisation of brain connectome data, used for graph metric analysis in this study is publicly available (https://github.com/RittmanResearch/maybrain). Code used for preprocessing and analysis in this manuscript is available at https://github.com/djw216/ucbj-fmri[92].

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

## Acknowledgements

We thank the participants, the Wolfson Brain Imaging Centre, and the staff at the Cambridge Centre for Parkinson-plus. We thank UCB Pharma, for providing the precursor for [$^{11}$C]UCB-J. This study was co-funded by the Wellcome Trust (103838 and 220258, JBR); the Cambridge Uni-versity Centre for Parkinson-Plus (RG95450, JBR); the National Institute for Health Research (NIHR) Cambridge Biomedical Research Centre (NIHR203312 relating to infrastructure funding: the views expressed are those of the authors and not necessarily those of the NIHR or the Department of Health and Social Care; Medical Research Council (MC_UU_00030/14; SUAG/092 G116768, JBR); Race Against Dementia Alzheimer's Research UK (ARUK-RADF2021A-010, MM); and the Asso-ciation of British Neurologists, Patrick Berthoud Charitable Trust (RG99368, NH).

## Author contributions

D.J.W., J.O'B. and J.B.R. had full access to the study data and took responsibility for the integrity of the data and accuracy of analysis. D.J.W., N.H., K.A.T., M.M., T.R., S.P.J., F.I.A., J.O'B. and J.B.R were responsible for concept development and design. N.H., M.M., G.S., M.N. and M.A.R. recruited patients from the Cambridge site and were responsible for data collection. K.P.B. and E.Mu. identified patients from UCL. T.D.F. and Y.T.H. were responsible for PET data acquisition, image reconstruction, image preprocessing, and determination of [$^{11}$C]UCB-J BP$_{ND}$ values and grey matter volumes. E. Ma. processed the diffusion sequences. D.J.W. processed other neuroimaging data and performed statistical analyses. D.J.W. drafted the manuscript. All authors critically reviewed the manuscript. J.B.R and J.O'B supervised the research and obtained funding.

## Competing interests

The authors declare the following competing interests: James B Rowe is a non-remunerated trustee of the Guarantors of Brain. He provides consultancy to Asceneuron, Astronautx, Astex, Biogen, Curasen, CumulusNeuro, Prevail, SVHealth and Wave, and has research grants from AZ-Medimmune, Janssen, Lilly and GSK as industry partners in the Dementias Platform UK. John T O'Brien has acted as a consultant for TauRx, Novo Nordisk, Biogen, Roche, Lilly and GE Healthcare and received grant or academic support from Avid/ Lilly, Merck, UCB and Alliance Medical. All other authors declare no competing interests.
