## [Peer Review File · Nature Communications]

Synaptic density affects clinical severity via network dysfunction in syndromes associated with Frontotemporal Lobar DegenerationREVIEWER COMMENTS

Reviewer #1 (Remarks to the Author):

The submitted manuscript describes a cross-sectional neuroimaging study of participants with FTD syndromes (bvFTD, PSP, and CBC) with the primary analysis focused on the associations between synaptic density measured with UCB-J PET and functional connectivity as determined by “weighted degree” using resting state BOLD. The authors hypothesize that synaptic loss from FTLD will “impair local and long-range functional connectivity.” One strength of the paper is the inclusion of heterogenous syndromes that represent likely cases of tauopathy and TDP-43 pathology. This sample should contain spatially distinct patterns of neurodegeneration that correspond to clinical/cognitive syndromes. The findings from these primary analyses of spatial relationships between connectivity and synaptic density across and withing disease phenotypes if of great interest. The additional analyses that include exploration of neuroreceptor/transporter distributions and LC integrity are of potential interest but are not described clearly. The results of this study are of interest since they explore the meaning of synaptic density measured with PET and help determine how this measure related to brain connectivity. This will be of relevance in all neurodegenerative disease research. There are a number of clarifications that need to be made to strengthen the quality of the manuscript.

1. The introduction states, “To examine syndrome-specific differences in the effect of synaptic loss, we reduced data dimensionality by independent component analysis. We predicted that reduced synaptic density would be associated with connectivity loss of the same region and the region to which it is connected, in such a way as relates to individual differences in cognition.” The prediction here is not clearly stated and needs clarification. If it is more exploratory, it would be helpful to describe the analyses in the results and discussion, but not state as a main focus in the introduction.

2. Results Line 102. I don’t think that the corrected differences tell the entire story regarding differences that are biologically plausible. I would suggest describing some of those group differences in the results section, but stating that they were not significant with correction for multiple comparisons. The format used with weighted degree would seem appropriate.

3. For PCA, images were warped to a common space non-linearly and resliced. Wouldn’t this eliminate the PVC and would increase the bias due to PVE in PCA analyses and any analyses that used these non-linearly resliced images?

4. I am struggling to follow the process for including neuroreceptor/transporter maps as a predictor of connectivity. The methods, results, and figures related to these analyses need clarification. Is the correlation done across regional data that is averaged for all participants? What is the sample size?

5. Line 533: References can be added here for how synaptic density measured with UCB-J PET has been associated with clinical severity.

Malpetti M, Jones PS, Cope TE, Holland N, Naessens M, Rouse MA, Rittman T, Savulich G, Whiteside DJ, Street D, Fryer TD, Hong YT, Milicevic Sephton S, Aigbirhio FI, O'Brien JT, Rowe JB. Synaptic Loss in Frontotemporal Dementia Revealed by [11 C]UCB-J Positron Emission Tomography. *Ann Neurol*. 2023 Jan;93(1):142-154. doi: 10.1002/ana.26543. Epub 2022 Nov 16. PMID: 36321699; PMCID: PMC10099663.

Mecca AP, O'Dell RS, Sharp ES, Banks ER, Bartlett HH, Zhao W, Lipior S, Diepenbrock NG, Chen MK, Naganawa M, Toyonaga T, Nabulsi NB, Vander Wyk BC, Arnsten AFT, Huang Y, Carson RE, van Dyck CH. Synaptic density and cognitive performance in Alzheimer's disease: A PET imaging study with [11 C]UCB-J. *Alzheimers Dement*. 2022 Dec;18(12):2527-2536. doi: 10.1002/alz.12582. Epub 2022 Feb 17. PMID: 35174954; PMCID: PMC9381645.

Vanderlinden G, Ceccarini J, Vande Casteele T, Michiels L, Lemmens R, Triau E, Serdons K, Tournoy J, Koole M, Vandenbulcke M, Van Laere K. Spatial decrease of synaptic density in amnesic mild cognitive impairment follows the tau build-up pattern. *Mol Psychiatry*. 2022 Oct;27(10):4244-4251. doi: 10.1038/s41380-022-01672-x. Epub 2022 Jul 6. PMID: 35794185.

6. For figure 1b, line 127 states, "In patients, assessing the [11C]UCB-J BPND effect within region showed stronger weighted degree-[11C]UCB-J BPND relationships in temporal, parietal, cingulate and superior frontal regions (Fig. 1b)." Stronger compared to what? It would help if thresholding were applied to the regional figure to mask weak correlations. Are the negative correlations meaningful, are any strong?

7. Figure 2b should be thresholded based on some significance level to help understand the strength of these contributions.

8. The LC analysis depicted in Figure 2c is not clear. What participants were included in this analysis and what were the diagnoses? What were the dependent and independent variables for this model. The figure axes titles should be clear based on the legend.

9. In the discussion, "In our example case, we have found that estimates of noradrenaline contribution to connectivity correspond to direct assessment of the locus coeruleus noradrenergic system, using a neuromelanin-sensitive sequence at 7T. Our study therefore provides important mechanistic insights into the multiple and interacting pathogenic processes of FTLD that result in diverse clinical syndromes." Without further explanation, these analyses and findings don't seem to contribute to the main message of this manuscript and appear tangential.

10. It is not clear in the analysis how this hypothesis is supported, "third, brain regions implicated in cognitive symptoms in neurodegeneration can have an effect remote from atrophy or synaptic loss." How is it clear that changes in connectivity occur remotely from areas of synaptic loss?

11. Some additional minor comments:

- a. Indicate sample sizes in all figures.
- b. What is meant by the SA subscript when reporting P values?
- c. NAT is not defined (assume noradrenaline transporter)

Reviewer #2 (Remarks to the Author):

This is a technically solid and analytically sophisticated study of the relationship between synaptic density, measured with UCB-J PET, functional connectivity, measured with resting state MRI, and clinical severity. The authors conclude that synaptic loss drives abnormal connectivity, both locally and remotely, and that behavioral abnormalities are the consequence of this cascade of dysfunction. This is a highly credible conclusion. So credible, in fact, that it would have been quite surprising for the authors to find something that contradicts this conclusion. I submit that what the authors present as the overarching hypothesis (lines 53-55) and specific hypotheses (lines 284-290) motivating this study are not hypotheses but generally well established facts about neurodegenerative disorders. The authors have done an impressive job of pulling together a number of methods and analyses but their conclusions – that synaptic loss/dysfunction is a better predictor than atrophy, that damage to one area of the brain produces remote effects, and that breakdown of the interaction of brain networks drives clinical dysfunction – are not novel. Have the authors presented any new concepts? Have they presented a new biomarker? A new way of stratifying research populations? Have they identified useful subgroups for future research?

The rationale for and utility of adding the neurotransmitter receptor – transporter map analysis isn't clear. This looks simply like a way of reducing some of the variance in their analyses without clear biologic rationale. Is this just a way of trying to track more synaptic loss/dysfunction? Sending coals to Newcastle? The great majority of CNS synapses are glutamatergic with ionotropic receptors, for which we don't have adequate tracers. What is the significance of the results with D1 receptors or NE transporters? The authors seem to imply that NE neuron degeneration is a substantial factor in clinical features, but this seems tenuous.

Minor Points:

- 1) How do post-mortem studies identify early and extensive synaptic loss (line 27)?
- 2) How does the "pathogenic cascade" (line 50) include synaptic loss? Isn't the latter the consequence of pathogenic cascades?
- 3) How does PET imaging validate post-mortem studies (line 66)? Post-mortem work is the conventional "gold standard."
- 4) This is mainly a study of PSP subjects. Were they all Richardson syndrome – this should be made clear? Clinical information should include any treatments.
- 5) The finding of largest effects sizes in frontal lobes and basal ganglia is likely largely result of subject selection.
- 6) The Discussion is prolix.

Reviewer #3 (Remarks to the Author):

The authors present an innovative study of brain connectivity using multimodal imaging (synaptic density via PET, fMRI and in a subset n=10 7T MRI integrity of the locus coeruleus) in frontotemporal dementia patients to test relationships between synaptic loss and brain network degeneration and

clinical symptoms. They compared fMRI measure of weighted degree to synaptic density on PET and found an association of connectivity with synaptic density in FTD patients (but not controls) and regional patterns of association in frontotemporal regions of high synaptic loss. Next, they include public atlas neurotransmitter data from PET in healthy controls and find specific neurotransmitter PET uptake associations contribute to variance in connectivity in FTD syndromes (NE and 5HT1A in PSP subgroup). They supplement this analysis using 7T measure of LC integrity in 10 PSP patients and link decreasing LC integrity with stronger associations of NE/synaptic density on connectivity. Finally, they perform data reduction techniques from whole brain analysis to identify clusters of synaptic loss with functional connectivity in each FTD syndrome and relate these to cognitive and motor scales.

The authors conclude that lower synaptic density is associated with lower functional connectivity in the brain that is greater than predicted by grey matter volume, and that functional connectivity modulates the relationship between clinical severity and synaptic density. Moreover, they state their data supports the hypotheses that heterogenous proteinopathies result in overlapping clinical syndromes based on synaptic dysfunction in a single brain region, synaptic dysfunction can cause connectivity changes at distal sites with minimal atrophy and that brain regions implicated in cognitive symptoms can have an effect remote from atrophy or synaptic loss.

This is a unique study with novel multimodal imaging, particularly linking PET measure of synaptic density to brain connectivity with sophisticated image processing and computational methods. Overall, the methods appear rigorous and thoughtful for potential confounds, but enthusiasm is weakened somewhat due to the limitations in terms of sample size for specific subgroups of FTD with heterogenous underlying pathologies (PSP=29, CBS=16, bvFTD=8). While this is a rare dataset to have this deep multimodal imaging data, the lower n, particularly for the bvFTD group, somewhat limits the ability to support the broader claims above in terms of neurodegeneration and brain connectivity which affect the significance of the study as outlined below. Moreover, it is unclear if patterns observed here in FTD can generalize to other neurodegenerative dementias. I have other more specific detailed comments below: Additional analyses and clarifications are needed to fully support the claims of the authors summarized above. Most notably, it would be helpful to clearly demonstrate that synaptic density measured by PET is providing additional information not obtained from conventional structural imaging. It would be helpful to include the full statistical model in the supplement to demonstrate the effect size when adjusting for GM volume in Figure 1, and it would also be helpful to illustrate the regional patterns of GM volume loss in relationship with synapse loss to determine if there are regions that are GM intact but reduced synaptic tracer uptake to support this claim.

With lack of autopsy data they cannot directly justify the claim that their data “supports the hypotheses that heterogenous proteinopathies result in overlapping clinical syndromes from disease in single brain regions.” Moreover, there is relative large overlap in boxplots in Figure 3a for each syndrome suggesting relative overlap in clusters of brain regions with synaptic loss across syndromes which is contrary to this claim.

The authors also make the claim “synaptic dysfunction causes connectivity change at sites of minimal atrophy, potentially contributing to the behavioural prodrome in presymptomatic dementias.” This statement is speculative as there is no data for presymptomatic individuals presented.

Additional analyses are needed to support the claim “brain regions implicated in cognitive symptoms in neurodegeneration can have an effect remote from atrophy or synaptic loss” as there is considerable overlap in clusters of PET synaptic density (green) and fMRI connectivity (blue) in Figure 4 models. A

more direct analysis of areas of high synaptic loss in each syndrome with areas of reduced connectivity and domain specific clinical assessments (i.e. ACE with bvFTD, PSPRS with PSP subgroup) would be helpful to more strongly support this hypothesis.

The assessment of connectivity is based on a single network metric of weighted degree calculated from a custom parcellation of the brain to provide roughly equal sized brain regions for nodes. It would be helpful to demonstrate reproducibility of findings with a different parcellation method and/or complementary network statistic to obtain converging evidence.

Similarly, to increase the significance of the findings more broadly to the field of neuroscience beyond the FTD field, it would be helpful to test for relationships in other neurodegenerative diseases and/or use another imaging metric of synaptic density such as NODDI for converging evidence to support the main conclusions of the study.

A few additional minor comments below:

It would be helpful to include more details on how patient cohort was selected to conform to STROBE reporting guidelines, including the number of potentially eligible subjects and # excluded for various reasons.

A strength of the study is the use of amyloid PET to exclude CBS patients with likely primary AD pathology, but it is not clear if AD biomarkers were examined in the other syndromes, as AD co-pathology is not uncommon in older FTD patients and can influence brain connectivity and structural imaging in FTD.

It would be helpful to include more details of the timing of imaging acquisition and clinical assessments if there were time intervals between collection of MRI, PET or clinical assessments for patients or all in the same visit.

Response to reviewers

We thank the reviewers for their suggestions that improve our manuscript on the relationship between synaptic density, functional connectivity, and clinical severity in three syndromes associated with frontotemporal lobar degeneration (FTLD, associated with bvFTD, PSP, and CBS). We provide detail of our additional analyses and findings below in the point-by-point responses to the reviewers.

REVIEWER COMMENTS

Reviewer #1 (Remarks to the Author):

The submitted manuscript describes a cross-sectional neuroimaging study of participants with FTD syndromes (bvFTD, PSP, and CBS) with the primary analysis focused on the associations between synaptic density measured with UCB-J PET and functional connectivity as determined by “weighted degree” using resting state BOLD. The authors hypothesize that synaptic loss from FTLD will “impair local and long-range functional connectivity.” One strength of the paper is the inclusion of heterogeneous syndromes that represent likely cases of tauopathy and TDP-43 pathology. This sample should contain spatially distinct patterns of neurodegeneration that correspond to clinical/cognitive syndromes. The findings from these primary analyses of spatial relationships between connectivity and synaptic density across and within disease phenotypes is of great interest. The additional analyses that include exploration of neuroreceptor/transporter distributions and LC integrity are of potential interest but are not described clearly. The results of this study are of interest since they explore the meaning of synaptic density measured with PET and help determine how this measure related to brain connectivity. This will be of relevance in all neurodegenerative disease research. There are a number of clarifications that need to be made to strengthen the quality of the manuscript.

RESPONSE: We thank the reviewer for their constructive and thoughtful suggestions.

REVIEWER: 1. The introduction states, “To examine syndrome-specific differences in the effect of synaptic loss, we reduced data dimensionality by independent component analysis. We predicted that reduced synaptic density would be associated with connectivity loss of the same region and the region to which it is connected, in such a way as relates to individual differences in cognition.” The prediction here is not clearly stated and needs clarification. If it is more exploratory, it would be helpful to describe the analyses in the results and discussion, but not state as a main focus in the introduction.

RESPONSE: We have adjusted the introduction to provide greater clarity (page 4 paragraph 2):

To examine syndrome-specific differences in the effect of synaptic loss, we reduced data dimensionality by independent component analysis. We found that reduced synaptic density was associated with connectivity loss in such a way as moderates and explains additional variance in individual differences in cognition. We further explored the effect of synaptic loss on connectivity, distinguishing the effect of synaptic loss in a region versus the region to which it is connected.

REVIEWER: 2. Results Line 102. I don't think that the corrected differences tell the entire story regarding differences that are biologically plausible. I would suggest describing some of those group differences in the results section, but stating that they were not significant with correction for multiple comparisons. The format used with weighted degree would seem appropriate.

RESPONSE: We have adjusted the results section in line with the reviewer's suggestion (page 6 paragraph 1).

REVIEWER: 3. For PCA, images were warped to a common space non-linearly and resliced. Wouldn't this eliminate the PVC and would increase the bias due to PVE in PCA analyses and any analyses that used these non-linearly resliced images?

RESPONSE: Partial volume correction using the iterative Yang method was applied in subject space, i.e., the space within which the partial volume error occurred, primarily to reduce the differential effect of atrophy across the cohort. Spatial normalization – required to enable independent component analysis on a spatially co-registered basis – will not negate this differential correction for atrophy.

To ensure that the observed patterns of synaptic density derived using independent component analysis of [¹¹C]UCB-J BP_{ND} maps were not primarily driven by atrophy, we repeated our analysis using the residuals derived from spatial regression of [¹¹C]UCB-J BP_{ND} maps against grey matter segmentations. Components were matched to those obtained in the primary analysis with spatial cross-correlation. Participant loadings for the matched component were well correlated (mean Pearson's R 0.81, range 0.58-0.96). We found between-group differences in these components with a highly similar pattern of group differences as found using [¹¹C]UCB-J BP_{ND} maps without spatial regression of grey matter

tissue probability maps (Supplementary Fig. 6), suggesting that atrophy or partial volume effects are not key determinants of these results.

REVIEWER: 4. I am struggling to follow the process for including neuroreceptor/transporter maps as a predictor of connectivity. The methods, results, and figures related to these analyses need clarification. Is the correlation done across regional data that is averaged for all participants? What is the sample size?

RESPONSE: We have carefully considered the reviewer's uncertainty about this analysis, which accord with points raised by the other two reviewers. Factors determining functional connectivity in neurodegeneration include inflammation, pathological protein, white matter disease, neurotransmitter deficits, metabolism, and cell death. Identifying their relative contribution to connectivity could enable in vivo mechanistic testing of their importance for patient outcome.

Our original paper's analysis used publicly available neurotransmitter receptor/transporter maps to quantify factors beyond synaptic and grey matter loss that alter connectivity. It was an example of how one can combine modalities to derive insights into the causes of individual differences. However, this analysis was an extension of the primary focus of the manuscript, and not related to our principal hypotheses.

Given the additional material included in this revision, we have therefore removed the section related to neurotransmitter maps.

REVIEWER: 5. Line 533: References can be added here for how synaptic density measured with UCB-J PET has been associated with clinical severity.

Malpetti M, Jones PS, Cope TE, Holland N, Naessens M, Rouse MA, Rittman T, Savulich G, Whiteside DJ, Street D, Fryer TD, Hong YT, Milicevic Sephton S, Aigbirhio FI, O'Brien JT, Rowe JB. Synaptic Loss in Frontotemporal Dementia Revealed by [11 C]UCB-J Positron Emission Tomography. *Ann Neurol*. 2023 Jan;93(1):142-154. doi: 10.1002/ana.26543. Epub 2022 Nov 16. PMID: 36321699; PMCID: PMC10099663.

Mecca AP, O'Dell RS, Sharp ES, Banks ER, Bartlett HH, Zhao W, Lipior S, Diepenbrock NG,

Chen MK, Naganawa M, Toyonaga T, Nabulsi NB, Vander Wyk BC, Arnsten AFT, Huang Y, Carson RE, van Dyck CH. Synaptic density and cognitive performance in Alzheimer's disease: A PET imaging study with [¹¹C]UCB-J. *Alzheimers Dement*. 2022 Dec;18(12):2527-2536. doi: 10.1002/alz.12582. Epub 2022 Feb 17. PMID: 35174954; PMCID: PMC9381645.

Vanderlinden G, Ceccarini J, Vande Castele T, Michiels L, Lemmens R, Triau E, Serdons K, Tournoy J, Koole M, Vandenbulcke M, Van Laere K. Spatial decrease of synaptic density in amnesic mild cognitive impairment follows the tau build-up pattern. *Mol Psychiatry*. 2022 Oct;27(10):4244-4251. doi: 10.1038/s41380-022-01672-x. Epub 2022 Jul 6. PMID: 35794185.

RESPONSE: We agree and have included these references in the manuscript.

REVIEWER: 6. For figure 1b, line 127 states, "In patients, assessing the [¹¹C]UCB-J BP_{ND} effect within region showed stronger weighted degree-[¹¹C]UCB-J BP_{ND} relationships in temporal, parietal, cingulate and superior frontal regions (Fig. 1b)." Stronger compared to what? It would help if thresholding were applied to the regional figure to mask weak correlations. Are the negative correlations meaningful, are any strong?

RESPONSE: The mixed effects model that tests the relationship between weighted degree and [¹¹C]UCB-J BP_{ND} includes a random slope for [¹¹C]UCB-J BP_{ND} within region. As a result we can derive standardised estimates (betas) of the [¹¹C]UCB-J BP_{ND}-weighted degree relationship by region. These need to be considered together with the [¹¹C]UCB-J BP_{ND} fixed-effect to understand total regional effect of [¹¹C]UCB-J BP_{ND} on weighted degree.

To make this clear, we have adjusted Fig. 1b by the fixed effect and thresholded to show that there are not strong negative regional correlations between weighted degree and [¹¹C]UCB-J BP_{ND}. We have also adjusted the wording in the text referenced by the reviewer to clarify that these relationships are standardised estimates (page 7 paragraph 1).

REVIEWER: 7. Figure 2b should be thresholded based on some significance level to help understand the strength of these contributions.

8. The LC analysis depicted in Figure 2c is not clear. What participants were included in

this analysis and what were the diagnoses? What were the dependent and independent variables for this model. The figure axes titles should be clear based on the legend.

9. In the discussion, "In our example case, we have found that estimates of noradrenaline contribution to connectivity correspond to direct assessment of the locus coeruleus noradrenergic system, using a neuromelanin-sensitive sequence at 7T. Our study therefore provides important mechanistic insights into the multiple and interacting pathogenic processes of FTLD that result in diverse clinical syndromes." Without further explanation, these analyses and findings don't seem to contribute to the main message of this manuscript and appear tangential.

REPOSE (to points 7-9): As per our response to the reviewer's fourth point, we have removed the analysis using neurotransmitter receptor/transporter maps from the manuscript, including the original Figure 2 and referenced paragraph in the discussion.

REVIEWER: 10. It is not clear in the analysis how this hypothesis is supported, "third, brain regions implicated in cognitive symptoms in neurodegeneration can have an effect remote from atrophy or synaptic loss." How is it clear that changes in connectivity occur remotely from areas of synaptic loss?

RESPONSE: We are glad of the opportunity to clarify this issue. The voxelwise analysis takes spatially independent regions of variation in synaptic density and derives patterns of component-specific functional connectivity with these regions for each participant. We find associations in functional connectivity with participant [^{11}C]UCB-J BP_{ND} component loadings both within the component and remotely from it. This is most evident with the striatal [^{11}C]UCB-J BP_{ND} component 1, where for example we find associations in functional covariance across the cortex with participant [^{11}C]UCB-J BP_{ND} loadings (Figure 2b).

We have included additional analyses where we explicitly test the relationship between reductions in synaptic density remote from a region, and their consequences for cognition (see '*The effect of remote connectivity on UCB-J BP_{ND} and clinical severity*,' from page 12 paragraph 1, and related methods from page 27 paragraph 2). Further details of this analysis and the results are set out below in response to the third reviewer.

We agree that the original wording highlighted by the reviewer may have been read to imply causation, rather than associations. We have adjusted the discussion as follows:

'We have shown that connectivity improves prediction and moderates the effect of synaptic density on clinical severity, with functional connectivity loss both overlapping with and occurring remotely from sites of reduced synaptic density. These remote connectivity differences can themselves moderate the relationship between synaptic density and cognition.'

REVIEWER: 11. Some additional minor comments:

a. Indicate sample sizes in all figures.

RESPONSE: We have modified the figures and figure legends to include sample size.

REVIEWER: b. What is meant by the SA subscript when reporting P values?

RESPONSE: In the original manuscript the SA subscript referred to P values calculated using *spatial-autocorrelation* preserving permutation testing when performing regional comparisons for group averaged maps. We have removed this notation given the removal of the section on neurotransmitter receptor/transporter maps.

REVIEWER: c. NAT is not defined (assume noradrenaline transporter)

RESPONSE: Correct, but this acronym is no longer included in the manuscript.

Reviewer #2 (Remarks to the Author):

REVIEWER: This is a technically solid and analytically sophisticated study of the relationship between synaptic density, measured with UCB-J PET, functional connectivity, measured with resting state MRI, and clinical severity. The authors conclude that synaptic loss drives abnormal connectivity, both locally and remotely, and that behavioral abnormalities are the consequence of this cascade of dysfunction. This is a highly credible conclusion. So credible, in fact, that it would have been quite surprising for the authors to find something that contradicts this conclusion. I submit that what the authors present as the overarching hypothesis (lines 53-55) and specific hypotheses (lines 284-290) motivating this study are not hypotheses but generally well established facts about neurodegenerative disorders. The authors have done an impressive job of pulling together a number of methods and analyses but their conclusions – that synaptic

loss/dysfunction is a better predictor than atrophy, that damage to one area of the brain produces remote effects, and that breakdown of the interaction of brain networks drives clinical dysfunction – are not novel. Have the authors presented any new concepts? Have they presented a new biomarker? A new way of stratifying research populations? Have they identified useful subgroups for future research?

RESPONSE: We agree that preclinical studies and post-mortem work suggest that extensive synaptic loss occurs in neurodegenerative disease and is strongly associated with clinical severity. Preclinical work also supports the hypothesis that synaptic health and plasticity are integral in generating neurophysiological connections and therefore maintaining cognitive function.

However, much less has been established about these relationships in vivo. It is well recognised that cognitive function in neurodegenerative diseases is incompletely explained by pathological burden and atrophy. Measuring and testing the relationship between synaptic density, network function, and clinical severity allows us to narrow this explanatory gap in humans in vivo, and to model the mechanisms of human neurodegenerative disease.

We did not seek to establish any of the imaging metrics as clinical biomarkers. Rather, we demonstrate that resting state fMRI can be used to quantify the consequences of upstream neuropathological changes, which is useful for experimental and translational research but not as a clinical biomarker. Demonstrating the relationship between synaptic density and functional connectivity is important, given the importance of synaptic function for clinical severity. Furthermore our work shows: i) that [^{11}C]UCB-J BP_{ND} , which is a measure of synaptic density rather than function, is associated with functional change; and ii) that synaptic loss is an important factor in resting state functional connectivity in neurodegenerative diseases. This finding is of note given the uncertainty as to the cellular and molecular processes that give rise to the BOLD signal. The manuscript has been modified to clarify these motivations.

REVIEWER: The rationale for and utility of adding the neurotransmitter receptor – transporter map analysis isn't clear. This looks simply like a way of reducing some of the variance in their analyses without clear biologic rationale. Is this just a way of trying to track more synaptic loss/dysfunction? Sending coals to Newcastle? The great majority of CNS synapses are glutamatergic with ionotropic receptors, for which we don't have adequate tracers. What is the significance of the results with D1 receptors or NE

transporters? The authors seem to imply that NE neuron degeneration is a substantial factor in clinical features, but this seems tenuous.

RESPONSE: As per the response to the first reviewer, we have removed this section from the manuscript.

We disagree with the suggestion that changes to the noradrenergic system are not an important determinant of functional connectivity. There is evidence to the contrary in health (e.g. Shine et al., 10.1038/s41593-018-0312-0.) and disease (e.g. Rae et al, 10.1093/brain/aww138.). Moreover, noradrenergic dysfunction is an important and potentially reversible factor in clinical manifestations in FTLD-associated syndromes (reviewed by Holland et al, 10.1093/brain/awab111). However, this issue lies beyond the scope of the revised manuscript.

REVIEWER: Minor Points:

1) How do post-mortem studies identify early and extensive synaptic loss (line 27)?

RESPONSE: We have removed the word ‘early’ from the abstract, which referred to findings in preclinical models of neurodegenerative disease.

REVIEWER: 2) How does the “pathogenic cascade” (line 50) include synaptic loss? Isn’t the latter the consequence of pathogenic cascades?

RESPONSE: Given the array of proposed mechanisms that lead to clinical manifestation in neurodegenerative diseases, and that highly cited dynamic biomarker models of the pathological cascade in neurodegeneration include even clinical function (e.g. Figure 2, Jack et al 10.1016/S1474-4422(09)70299-6), we think the sentence is in keeping with the term’s common usage and that the meaning is clear and correct. The synaptic loss is a downstream consequence of several molecular and cellular processes (misfolded protein toxicity, inflammation, mitochondrial impairment and oxidative stress etc etc), but it lies upstream of the failures of neurotransmission, connectivity, cognition, and clinical outcome. We therefore suggest that synaptic loss is “mid-way” in the pathogenic cascade between root cause and clinical outcome in neurodegenerative disorders.

REVIEWER: 3) How does PET imaging validate post-mortem studies (line 66)? Post-mortem work is the conventional “gold standard.”

RESPONSE: Corroboration, accord or correlation would be better terms than validation in this context. We have amended this line as follows (page 3 paragraph 3):

Reductions in [¹¹C]UCB-J non-displaceable binding potential (BP_{ND}), a metric of binding site density, correlate with clinical severity and are observed in the distributions also found in post mortem studies.

REVIEWER: 4) This is mainly a study of PSP subjects. Were they all Richardson syndrome – this should be made clear? Clinical information should include any treatments.

RESPONSE: As per the original manuscript, all the PSP participants had Richardson’s syndrome (page 18 paragraph 1). Information on medications for participants with FTLD-associated syndromes is included in Supplementary Table 1.

REVIEWER: 5) The finding of largest effects sizes in frontal lobes and basal ganglia is likely largely result of subject selection.

RESPONSE: We agree - it is reassuring that the distribution of synaptic loss is in keeping with post-mortem studies and the expected effects of pathology in syndromes associated with frontotemporal lobar degeneration.

REVIEWER: 6) The Discussion is prolix.

RESPONSE: The discussion has been updated and shortened.

REVIEWER: Reviewer #3 (Remarks to the Author):

The authors present an innovative study of brain connectivity using multimodal imaging (synaptic density via PET, fMRI and in a subset n=10 7T MRI integrity of the locus coeruleus) in frontotemporal dementia patients to test relationships between synaptic

loss and brain network degeneration and clinical symptoms. They compared fMRI measure of weighted degree to synaptic density on PET and found an association of connectivity with synaptic density in FTD patients (but not controls) and regional patterns of association in frontotemporal regions of high synaptic loss. Next, they include public atlas neurotransmitter data from PET in healthy controls and find specific neurotransmitter PET uptake associations contribute to variance in connectivity in FTD syndromes (NE and 5HT1A in PSP subgroup). They supplement this analysis using 7T measure of LC integrity in 10 PSP patients and link decreasing LC integrity with stronger associations of NE/synaptic density on connectivity. Finally, they perform data reduction techniques from whole brain analysis to identify clusters of synaptic loss with functional connectivity in each FTD syndrome and relate these to cognitive and motor scales.

The authors conclude that lower synaptic density is associated with lower functional connectivity in the brain that is greater than predicted by grey matter volume, and that functional connectivity modulates the relationship between clinical severity and synaptic density. Moreover, they state their data supports the hypotheses that heterogenous proteinopathies result in overlapping clinical syndromes based on synaptic dysfunction in a single brain region, synaptic dysfunction can cause connectivity changes at distal sites with minimal atrophy and that brain regions implicated in cognitive symptoms can have an effect remote from atrophy or synaptic loss.

This is a unique study with novel multimodal imaging, particularly linking PET measure of synaptic density to brain connectivity with sophisticated image processing and computational methods. Overall, the methods appear rigorous and thoughtful for potential confounds, but enthusiasm is weakened somewhat due to the limitations in terms of sample size for specific subgroups of FTD with heterogenous underlying pathologies (PSP=29, CBS=16, bvFTD=8). While this is a rare dataset to have this deep multimodal imaging data, the lower n, particularly for the bvFTD group, somewhat limits the ability to support the broader claims above in terms of neurodegeneration and brain connectivity which affect the significance of the study as outlined below. Moreover, it is unclear if patterns observed here in FTD can generalize to other neurodegenerative dementias. I have other more specific detailed comments below:

RESPONSE: We thank the reviewer their overall assessment and recognition of what we also see as the key aspects of the paper.

We agree that quantifying synaptic health in neurodegenerative diseases using additional imaging metrics provides useful converging evidence for the importance of synapses in determining *in vivo* connectivity. We have therefore included analyses using the diffusion method NODDI, which can be used to derive indirect measures of synaptic health. We find that the NODDI derived metric Orientation Dispersion Index is related to weighted degree beyond volumetric measures, supporting the conclusions drawn using [¹¹C]UCB-J. These results are set out in the manuscript from page 7 paragraph 2, with further detail and figures also provided below in the responses to the reviewer.

Although the sample size of the individual syndromic groups are moderate, the effect size for differences in [¹¹C]UCB-J BP_{ND} when comparing patients to healthy controls are large, and indeed larger than volumetric measures (see supplementary figures 1-2). Our study is adequately powered to detect relationships between synaptic density and functional connectivity. We show in additional analyses that the relationship between weighted degree and [¹¹C]UCB-J BP_{ND} holds in each of the clinical syndromes as well as in the whole patient group (page 7 paragraph 1). We have expanded bvFTD from n=8 to n=10. Our focus was on syndromes associated with frontotemporal lobar degeneration, which are characterised by heterogeneous yet overlapping clinical and pathological features. Our work aims to provide insight into the importance of synaptic function in determining clinical convergence and heterogeneity in neurodegenerative diseases. BvFTD, PSP, and CBS represent a spectrum of diseases with distinct molecular pathologies but pathological and clinical commonalities. Studying distinct FTLD syndromes allows us to capture variations in anatomical distribution of synaptic loss and understand its consequence for connectivity and clinical severity, utilising clinical measures that are relevant and comparable across the FTLD-spectrum.

REVIEWER: Additional analyses and clarifications are needed to fully support the claims of the authors summarized above. Most notably, it would be helpful to clearly demonstrate that synaptic density measured by PET is providing additional information not obtained from conventional structural imaging. It would be helpful to include the full statistical model in the supplement to demonstrate the effect size when adjusting for GM volume in Figure 1, and it would also be helpful to illustrate the regional patterns of GM

volume loss in relationship with synapse loss to determine if there are regions that are GM intact but reduced synaptic tracer uptake to support this claim.

RESPONSE: We have performed new analyses and provided further details in the manuscript to make clear that [¹¹C]UCB-J BP_{ND} provides additional information beyond volumetric methods, both in showing differences from controls and in modelling functional connectivity. The full linear mixed-effects model for comparison between weighted degree and [¹¹C]UCB-J BP_{ND} with regional grey matter volume as a covariate of no interest is included in the supplementary materials (Supplementary Table 6).

Supplementary Table 6. Fixed effects for relationship between synaptic density and weighted degree including grey matter volume

	Std Beta	SE	T value	P
Intercept	-0.04	0.15	-0.30	0.76
UCB-J	0.18	0.03	6.2	1x10 ⁻⁹
Grey matter vol	0.12	0.03	3.9	0.0001
Age	-0.15	0.09	-1.6	0.12
Mean DVARS	0.16	0.09	1.7	0.10
Sex	0.07	0.19	0.37	0.71
Cortical/subcortical	0.02	0.09	0.16	0.87

DVARS the spatial standard deviation of successive difference images

In Supplementary Figure 1 we show the regional differences between patients and controls for all modalities, with age, sex and total intracranial volume (for volumetric measures) included as covariates. Effects sizes are larger for [¹¹C]UCB-J BP_{ND} than grey matter volume, with regions that show differences in [¹¹C]UCB-J BP_{ND} but not for grey matter volume.

b UCB-J

d GM volume

*Supplementary Figure 1: Regional differences between healthy controls and patients in **b**) [^{11}C]UCB-J BP_{ND} , and **d**) grey matter volume. All parcels shown are significantly different after FDR-correction for multiple comparisons across regions.*

We also demonstrate that regional/voxelwise differences in [^{11}C]UCB-J BP_{ND} exist between patients and controls with regional grey matter volumes/ grey matter segmentations included as a regional/voxelwise regressors.

*Supplementary Figure 2: **a**) Regional and **b**) voxelwise differences between healthy controls and patients in [^{11}C]UCB-J BP_{ND} with grey matter volumes/ brainstem total volumes included as a covariate of no interest. All regional parcels shown are significantly different after FDR-correction for multiple comparisons across regions. Voxelwise statistics calculated using 1000 permutations within a grey matter mask. FEW family-wise error.*

We further compare the relationship between regional [^{11}C]UCB-J BP_{ND} and weighted degree across all participants, with regional grey matter volumes as a covariate of no interest in addition to age, sex, fMRI motion, and total intracranial volume. We find [^{11}C]UCB-J BP_{ND} explains regional weighted degree beyond grey matter volume.

Uncorrected $p < 0.05$

FDR $p < 0.05$

Supplementary Figure 3: Regional relationship between weighted degree and [^{11}C]UCB-J BP_{ND} with regional grey matter volume as a covariate of no interest in addition to age, sex, fMRI motion, and total intracranial volume.

REVIEWER: With lack of autopsy data they cannot directly justify the claim that their data “supports the hypotheses that heterogenous proteinopathies result in overlapping clinical syndromes from disease in single brain regions.” Moreover, there is relative large overlap in boxplots in Figure 3a for each syndrome suggesting relative overlap in clusters of brain regions with synaptic loss across syndromes which is contrary to this claim.

RESPONSE: In the updated discussion section we have removed this sentence, as it is not necessary to the principal hypotheses.

REVIEWER: The authors also make the claim “synaptic dysfunction causes connectivity change at sites of minimal atrophy, potentially contributing to the behavioural prodrome in presymptomatic dementias.” This statement is speculative as there is no data for presymptomatic individuals presented.

RESPONSE: Synaptic loss in presymptomatic adults at genetic risk of FTD has been published previously, even though not in our study (eg. Malpetti et al, doi: 10.1002/acn3.51407; with similar results from the MINDMAPS consortium). Nonetheless we have removed the reference to presymptomatic participants, as they were not in our current study.

REVIEWER: Additional analyses are needed to support the claim “brain regions implicated in cognitive symptoms in neurodegeneration can have an effect remote from atrophy or synaptic loss” as there is considerable overlap in clusters of PET synaptic

density (green) and fMRI connectivity (blue) in Figure 4 models. A more direct analysis of areas of high synaptic loss in each syndrome with areas of reduced connectivity and domain specific clinical assessments (i.e. ACE with bvFTD, PSPRS with PSP subgroup) would be helpful to more strongly support this hypothesis.

RESPONSE: We have extended our analyses to test more precisely the relationship between reductions in synaptic density remote from a region. The results from this analysis form a new section of the manuscript, titled '*The effect of remote connectivity on clinical severity*' (page 12 para 1). The related methods are start at page 27 paragraph 2 (titled '*Remote connectivity differences*').

For this analysis we consider fMRI connectivity scores both within a thresholded [¹¹C]UCB-J BP_{ND} component mask and outside the mask but within regions that control participants are connected to. We repeated the model selection process using connectivity scores outside the mask, and find as follows (page 12 paragraph 1):

We repeated the model selection process above using connectivity scores outside the [¹¹C]UCB-J BP_{ND} mask. We found that for the final model predicting ACE-R (Supplementary Table 9), the fMRI-[¹¹C]UCB-J BP_{ND} interaction was significant for component 4 for the ACE-R (Component 4 Interaction Std Beta 0.28, P=0.005). The final model predicting PSPRS included only [¹¹C]UCB-J BP_{ND} component 1 (Supplementary Table 10), although the fMRI-[¹¹C]UCB-J BP_{ND} interaction was significant for component 5 with PSPRS as dependent variable, again using component scores outside the [¹¹C]UCB-J BP_{ND} component mask (Interaction Std Beta -0.31, P=0.006).

This supports the claim that remote but connected brain regions can moderate the effect of [¹¹C]UCB-J BP_{ND} on cognition. Given these findings we have modified the relevant section of the discussion as set out in response the first reviewer's tenth point.

REVIEWER: The assessment of connectivity is based on a single network metric of weighted degree calculated from a custom parcellation of the brain to provide roughly equal sized brain regions for nodes. It would be helpful to demonstrate reproducibility of findings with a different parcellation method and/or complementary network statistic to obtain converging evidence.

RESPONSE: To ensure the relationship between weighted degree and [¹¹C]UCB-J BP_{ND} is independent of parcellation, we considered both the non-reparcelled Hammersmith Atlas and the Brainnetome Atlas. For the full Hammersmith Atlas we found that our results were replicated (supplementary results).

*Weighted degree was associated with [¹¹C]UCB-J BP_{ND} in patients (Standardised Beta 0.19, $P=1 \times 10^{-11}$) but not in control participants (Standardised Beta 0.0 $P=0.90$). The group-by-¹¹C]UCB-J BP_{ND} interaction in a refitted model with all participants was significant (¹¹C]UCB-J BP_{ND}*Group Standardised Beta 0.076, $P=9 \times 10^{-5}$). The relationship between [¹¹C]UCB-J BP_{ND} and weighted degree was also observed in each FTLN syndrome individually (bvFTD Standardised Beta 0.19 $P=3 \times 10^{-5}$; CBS Standardised Beta 0.14 $P=0.002$; PSP Standardised Beta 0.25, $P<9 \times 10^{-11}$).*

Using the Brainnetome Atlas, we again found that the relationship between weighted degree and [¹¹C]UCB-J BP_{ND} was stronger in patients than controls, and observed in each FTLN syndrome individually, as set out in the supplement.

*We fit the same linear mixed-effects models using the Brainnetome parcellation. Weighted degree was associated with [¹¹C]UCB-J BP_{ND} in patients (Standardised Beta 0.14, $P<2 \times 10^{-16}$) and also in control participants (Standardised Beta 0.05 $P=0.004$). The group-by-¹¹C]UCB-J BP_{ND} interaction in a refitted model with all participants was significant (¹¹C]UCB-J BP_{ND}*Group Standardised Beta 0.037, $P=0.002$). The relationship between [¹¹C]UCB-J BP_{ND} and weighted degree was also observed in each FTLN syndrome individually (bvFTD Standardised Beta 0.17 $P<2 \times 10^{-165}$; CBS Standardised Beta 0.12 $P=1 \times 10^{-11}$; PSP Standardised Beta 0.15, $P<2 \times 10^{-16}$).*

We also note that the relationship between [¹¹C]UCB-J BP_{ND} and connectivity is shown not only with weighted degree, but also in our voxel-wise analysis using [¹¹C]UCB-J BP_{ND} components as templates for regression. In conclusion, our results generalise to both non-parcellated data and parcellated data using multiple parcellations.

REVIEWER: Similarly, to increase the significance of the findings more broadly to the field of neuroscience beyond the FTD field, it would be helpful to test for relationships in other neurodegenerative diseases and/or use another imaging metric of synaptic density such as NODDI for converging evidence to support the main conclusions of the study.

RESPONSE: We thank the reviewer for this suggestion. Note our study includes three clinical disorders – FTD, PSP and CBS.

We have included further analysis using the diffusion MRI technique NODDI for the participants (75 of 79) who had undergone an appropriate sequence at the same scanning session as their echo planar imaging was obtained. We derived the NODDI metric

Orientation Dispersion Index (ODI) for parcels of the Hammersmith Atlas. We found (page 7 paragraph 2) that regional [¹¹C]UCB-J BP_{ND} values were related to regional ODI in both patients and controls. ODI values were significantly associated with regional weighted degree in patients but not in controls. The group-by ODI interaction was not significant. The relationship between ODI and weighted degree persisted with grey matter volumes and total intracranial volume as covariates.

In addition we assessed the relationship between weighted degree and [¹¹C]UCB-J BP_{ND}, ODI, and grey matter volume across participants. We found that the measures of synaptic health, [¹¹C]UCB-J BP_{ND}, ODI, were significantly related to weighted degree in more regions than grey matter volumes. Combining all three modalities using non-parametric combination testing further improved prediction of weighted degree. These results are illustrated in Figure 1c.

Figure 1c) Regions with a relationship with weighted degree in a general linear model for all participants for [¹¹C]UCB-J BP_{ND}, orientation dispersion index (ODI), grey matter volumes, and for the three modalities combined using non-parametric combination testing with correction across contrasts. Regions shown are significant with FDR-adjusted p-values < 0.05.

These results provide converging *in vivo* evidence that disruption to synaptic health in neurodegenerative diseases results in loss of functional connectivity, and that quantifying disruption to synapses provides additional information beyond volumetric T1 imaging.

REVIEWER: A few additional minor comments below:

It would be helpful to include more details on how patient cohort was selected to

conform to STROBE reporting guidelines, including the number of potentially eligible subjects and # excluded for various reasons.

RESPONSE: We have updated the methods section with information on the screening protocol and excluded participants.

Participants were initially screened via telephone, using the following exclusion criteria: a current or recent history of cancer within the last 5 years, concurrent use of the medication levetiracetam, any contraindications to undergoing an MRI, a history of ischaemic or haemorrhagic stroke evident on the MRI from the clinic, and any severe physical illness or co-morbidity that could limit their ability to fully participate in the study. Twenty-three participants who passed initial screening were not included in this study due to (i) positive Alzheimer's biomarkers (all CBS, n=9) or (ii) due to failure to complete all scanning sessions (n=14: Control n=8, bvFTD n=1, CBS n=1, PSP n=4).

REVIEWER: A strength of the study is the use of amyloid PET to exclude CBS patients with likely primary AD pathology, but it is not clear if AD biomarkers were examined in the other syndromes, as AD co-pathology is not uncommon in older FTD patients and can influence brain connectivity and structural imaging in FTD.

RESPONSE: We do not have results for Alzheimer's Disease biomarkers for the bvFTD or PSP cohort. However, for bvFTD and PSP, clinicopathological correlations are high in that these are rarely misdiagnosed from AD-pathology. We have extended the limitations section of the discussion to reference this point (page 16 paragraph 2).

REVIEWER: It would be helpful to include more details of the timing of imaging acquisition and clinical assessments if there were time intervals between collection of MRI, PET or clinical assessments for patients or all in the same visit.

RESPONSE: Imaging data in all participants was collected in two scanning sessions, one for synaptic PET imaging and a separate session with 3-Tesla MRI for echo-planar imaging sequences, volumetric imaging, and diffusion sequences. Participants with corticobasal syndrome completed a third session to determine their amyloid status. Median time between obtaining synaptic PET and 3-Tesla imaging was 58 days in patients (inter-quartile range 13-180 days) and 194 days in control participants (inter-quartile range 32-284 days). We have

updated the manuscript to include this information (page 19 paragraph 1). We also include scanning interval as a covariate in the linear mixed-effects model when testing for difference between patients and control participants in the weighted degree- ^{11}C UCB-J BP_{ND} relationship (page 7 paragraph 1), with no change to significance or effect size.

REVIEWERS' COMMENTS

Reviewer #1 (Remarks to the Author):

My concerns were addressed in this revision and I recommend this manuscript be accepted for publication in the current form.

Reviewer #2 (Remarks to the Author):

The authors have been responsive to my critique with the exception of my initial point about the significance of this work. I appreciate their polite response. I remain unconvinced that their basic conclusion has much novelty, though I have to acknowledge that the authors are expressing a legitimate opinion. I am compelled to add that the concept that focal pathology produces widespread, circuit-based consequences was pioneered in the 19th century.

Reviewer #3 (Remarks to the Author):

The authors have been extremely responsive to the last round of comments, including additional NODDI sequence for converging evidence of their findings and clearly showing an advantage over conventional structural imaging which is a key finding of the paper. I have no additional comments.

Response to reviewers

Reviewer #1 (Remarks to the Author):

My concerns were addressed in this revision and I recommend this manuscript be accepted for publication in the current form.

RESPONSE: We thank the reviewer for their thoughtful and constructive previous review and attention to this revision.

Reviewer #2 (Remarks to the Author):

The authors have been responsive to my critique with the exception of my initial point about the significance of this work. I appreciate their polite response. I remain unconvinced that their basic conclusion has much novelty, though I have to acknowledge that the authors are expressing a legitimate opinion. I am compelled to add that the concept that focal pathology produces widespread, circuit-based consequences was pioneered in the 19th century.

RESPONSE: We thank the reviewer for their response. In respect to the reviewer's reference to 19th century studies of localisation, we have updated the opening of our discussion to highlight how the concept of diaschisis has evolved from its initial formulation by von Monakow. Our findings demonstrate how functional or connectomal diaschisis (Carrera and Tononi, 2014, doi: 10.1093/brain/awu101) may arise. Our manuscript includes the following: (page 13 paragraph 1)

The corresponding reductions in functional connectivity are observed both at the site of the synaptic loss and remotely from it: as one of the potential mechanisms underlying diaschisis, functional diaschisis, and the recent formalisation of connectomal diaschisis.²⁵

Reviewer #3 (Remarks to the Author):

The authors have been extremely responsive to the last round of comments, including additional NODDI sequence for converging evidence of their findings and clearly showing an advantage over conventional structural imaging which is a key finding of the paper. I have no additional comments

REPOSENSE: We thank the reviewer for their time and for supporting publication of our manuscript.

In addition to these points in preparation for this revision we became aware that the null distribution for the statistical association between mean z-scored weighted degree and mean z-scored [¹¹C]UCB-J BP_{ND} was determined in our function by assessing z-scored [¹¹C]UCB-J BP_{ND} against [¹¹C]UCB-J BP_{ND} spatial-autocorrelation preserving null models.

More standard practice would be to generate the null distribution by taking the correlation between mean z-scored [^{13}C]UCB-J BP_{ND} with weighted degree spatial-autocorrelation preserving null models. As a result there is some change to the P values on page 6 paragraph 2, but not their significance and there are no changes to the interpretation of results.